# Quasi-Monte Carlo Graph Random Features

**Isaac Reid**
University of Cambridge
ir337@cam.ac.uk

**Krzysztof Choromanski**[*]
Google DeepMind
Columbia University
kchoro@google.com

**Adrian Weller**
University of Cambridge
Alan Turing Institute
aw665@cam.ac.uk

## Abstract

We present a novel mechanism to improve the accuracy of the recently-introduced class of *graph random features* (GRFs) [Choromanski, 2023]. Our method induces negative correlations between the lengths of the algorithm's random walks by imposing *antithetic termination*: a procedure to sample more diverse random walks which may be of independent interest. It has a trivial drop-in implementation. We derive strong theoretical guarantees on the properties of these *quasi-Monte Carlo GRFs* (q-GRFs), proving that they yield lower-variance estimators of the 2-regularised Laplacian kernel under mild conditions. Remarkably, our results hold for any graph topology. We demonstrate empirical accuracy improvements on a variety of tasks including a new practical application: time-efficient approximation of the graph diffusion process. To our knowledge, q-GRFs constitute the first rigorously studied quasi-Monte Carlo scheme for kernels defined on combinatorial objects, inviting new research on correlations between graph random walks.[1]

## 1 Introduction and related work

Kernel methods are ubiquitous in machine learning [Canu and Smola, 2006, Smola and Schölkopf, 2002, Kontorovich et al., 2008, Campbell, 2002]. Via the kernel trick, they provide a mathematically principled and elegant way to perform nonlinear inference using linear learning algorithms. The positive definite *kernel function* $k : \mathcal{X} \times \mathcal{X} \to \mathbb{R}$, defined on an input domain $\mathcal{X}$, measures the 'similarity' between two datapoints. Examples in Euclidean space include the Gaussian, linear, Matérn, angular and arc-cosine kernels [Williams and Rasmussen, 2006, Cho and Saul, 2011].

Though very effective on small datasets, kernel methods suffer from poor scalability. The need to materialise and invert the *kernel matrix* typically leads to a time-complexity cubic in the size of the dataset. Substantial research has been dedicated to improving scalability by approximating this matrix, notably including *random features* (RFs) [Rahimi and Recht, 2007, 2008, Avron et al., 2017, Liu et al., 2022] . These randomised mappings $\phi : \mathbb{R}^d \to \mathbb{R}^s$ construct low-dimensional feature vectors whose dot product equals the kernel evaluation in expectation:

$$k(\boldsymbol{x}, \boldsymbol{y}) = \mathbb{E} \left( \phi(\boldsymbol{x})^\top \phi(\boldsymbol{y}) \right). \tag{1}$$

This permits a low-rank decomposition of the kernel matrix which enables better time- and space-complexity than exact kernel methods. Random feature methods exist for a variety of Euclidean kernels with properties engineered for the desiderata and symmetries of the particular kernel being

---

[*]Senior lead.

[1]Source code is available at https://github.com/isaac-reid/antithetic_termination.

37th Conference on Neural Information Processing Systems (NeurIPS 2023).

approximated [Dasgupta et al., 2010, Johnson, 1984, Choromanski et al., 2020, Goemans and Williamson, 2001, Rahimi and Recht, 2007].

Kernels can also be defined on discrete input spaces such as *graphs*, which are the natural way to represent data characterised by local relationships (e.g. social networks or interacting chemicals [Albert and Barabási, 2002]) or when data is restricted to a lower-dimensional manifold than the original space [Roweis and Saul, 2000, Belkin and Niyogi, 2003]. We consider *graph kernels* $k : \mathcal{N} \times \mathcal{N} \to \mathbb{R}$ on the set of nodes $\mathcal{N}$ of a graph G. Examples include the diffusion, regularised Laplacian, $p$-step random walk and cosine kernels [Smola and Kondor, 2003, Kondor and Lafferty, 2002, Chung and Yau, 1999], which have found applications including in bioinformatics [Borgwardt et al., 2005], community detection [Kloster and Gleich, 2014] and recommender systems [Yajima, 2006]. More recently, these kernels have been used in manifold learning for deep generative modelling [Zhou et al., 2020] and for solving shortest path problems [Crane et al., 2017]. Substantial research effort has also been devoted to developing and analysing graph kernels $k : \mathcal{G} \times \mathcal{G} \to \mathbb{R}$, now taking entire graphs G $\in \mathcal{G}$ from graph spaces $\mathcal{G}$ as inputs rather than their nodes [Shervashidze et al., 2009, Vishwanathan et al., 2006, Shervashidze and Borgwardt, 2009], but we stress that these are not the subject of this paper.

The problem of poor kernel scalability is exacerbated in the graph domain because even computing the kernel matrix $\mathbf{K}$ is typically of at least cubic time-complexity in the number of nodes $N$. In contrast to kernels defined on points in $\mathbb{R}^d$, random feature methods for fixed graph kernels (c.f. kernel learning [Fang et al., 2021]) have proved challenging to construct. Only recently has a viable *graph random feature* (GRF) mechanism been proposed, which uses a series of random walkers depositing 'load' at each node they pass through [Choromanski, 2023]. GRFs provide an unbiased estimate of the matrix $(\mathbf{I}_N - \mathbf{U})^{-d}$, where $\mathbf{U}$ is a weighted adjacency matrix of the graph and $d \in \mathbb{N}$. The decomposition supports subquadratic time-complexity (again with respect to the number of nodes $N$) in downstream algorithms applying regularised Laplacian kernels. Moreover, the computation of GRFs admits a simple distributed algorithm that can be applied if a large graph needs to be split across machines. The author demonstrates the strong empirical performance of GRFs in speed tests, Frobenius relative error analysis and a $k$-means graph clustering task.

In the Euclidean setting, significant research has been dedicated to developing *quasi-Monte Carlo* (QMC) variants to RF methods that enjoy better convergence properties [Yang et al., 2014, Lyu, 2017, Dick et al., 2013]. By using correlated ensembles rather than i.i.d. random variables in the feature maps, one can suppress the mean squared error (MSE) of the kernel estimator. For example, *orthogonal random features* (ORFs) [Yu et al., 2016, Choromanski et al., 2020] improve the quality of approximation of the Gaussian kernel when using trigonometric or positive random features, and of the linear kernel in the *orthogonal Johnson-Lindenstrauss transformation* [Choromanski et al., 2017]. With positive random features, the recently-introduced class of *simplex random features* (SimRFs) performs even better [Reid et al., 2023]. This has been used to great effect in estimating the attention mechanism of Transformers [Vaswani et al., 2017], overcoming its prohibitive quadratic time-complexity scaling with token sequence length.

This invites the central question of this work: **how can we implement a QMC mechanism for random walks on a graph?** What do we mean by a 'diverse' sample in this context? Choromanski [2023] first identified the challenge of constructing *quasi-Monte Carlo GRFs* (q-GRFs). They suggested a high-level approach of reinforced random walks, but left to future work its theoretical and empirical analysis. In this paper, we provide a first concrete implementation of q-GRFs, proposing an unbiased scheme that correlates the length of random walks by imposing *antithetic termination*. We derive strong theoretical guarantees on the properties of this new class, proving that the correlations reduce the variance of estimators of the 2-regularised Laplacian kernel under mild conditions. Our results hold for any graph topology. We also demonstrate empirical accuracy improvements on a variety of tasks. We hope our new algorithm (hereafter referred to as 'q-GRFs' for brevity) will spur further research on correlations between graph random walks in machine learning.

We emphasise that, although we present antithetic termination through the lens of q-GRFs (an analytically tractable and important use case), it is fundamentally a procedure to obtain a more diverse ensemble of graph random walks and may be of independent interest. To illustrate this, in App 8.1 we show how it can alternatively be used to improve approximations of the PageRank vector [Page et al., 1998]. It could also be relevant e.g. for estimating graphlet statistics for kernels between graphs

[Chen et al., 2016, Ribeiro et al., 2021] or in some GNN architectures [Nikolentzos and Vazirgiannis, 2020].

The remainder of the manuscript is organised as follows. In **Sec. 2** we introduce the mathematical concepts and existing algorithms to be used in the paper, including the $d$-regularised Laplacian and diffusion kernels and the GRF mechanism. **Sec. 3** presents our novel q-GRFs mechanism and discusses its strong theoretical guarantees – in particular, that it provides lower kernel estimator variance than its regular predecessor (GRFs) under mild conditions. We provide a brief proof-sketch for intuition but defer full technical details to App. 8.4. We conduct an exhaustive set of experiments in **Sec. 4** to compare q-GRFs to GRFs, including: (a) quality of kernel approximation via computation of the relative Frobenius norm; (b) simulation of the graph diffusion process; (c) kernelised $k$-means node clustering; and (d) kernel regression for node attribute prediction. q-GRFs nearly always perform better and in some applications the difference is substantial.

## 2   Graph kernels and GRFs

### 2.1   The Laplacian, heat kernels and diffusion on graphs

An undirected, weighted graph $G(\mathcal{N}, \mathcal{E})$ is defined by a set of vertices $\mathcal{N} := \{1, ..., N\}$ and a set of edges $\mathcal{E}$ given by the unordered vertex pairs $(i, j) \in \mathcal{E}$ where $i$ and $j$ are neighbours, themselves associated with weights $W_{ij} \in \mathbb{R}$. The *weighted adjacency matrix* $\mathbf{W} \in \mathbb{R}^{N \times N}$ has matrix elements $W_{ij}$: that is, the associated edge weights if $(i, j) \in \mathcal{E}$ and 0 otherwise. Let $\mathcal{N}(i) := \{j \in \mathcal{N} | (i, j) \in \mathcal{E}\}$ denote the set of neighbours of node $i$.

Denote by $\mathbf{D} \in \mathbb{R}^{N \times N}$ the diagonal matrix with elements $D_{ii} := \sum_j W_{ij}$, the sum of edge weights connecting a vertex $i$ to its neighbours. The *Laplacian* of $G$ is then defined $\mathbf{L} := \mathbf{D} - \mathbf{W}$. The *normalised Laplacian* is $\widetilde{\mathbf{L}} := \mathbf{D}^{-\frac{1}{2}} \mathbf{L} \mathbf{D}^{-\frac{1}{2}}$, which rescales $\mathbf{L}$ by the (weighted) number of edges per node. $\mathbf{L}$ and $\widetilde{\mathbf{L}}$ share eigenvectors in the case of an equal-weights $d$-regular graph and play a central role in spectral graph theory; their analytic properties are well-understood [Chung, 1997].

In classical physics, diffusion through continuous media is described by the equation

$$\frac{d\mathbf{u}}{dt} = \nabla^2 \mathbf{u} \tag{2}$$

where $\nabla^2 = \frac{\partial^2}{\partial x_1^2} + \frac{\partial^2}{\partial x_2^2} + ... \frac{\partial^2}{\partial x_N^2}$ is the *Laplacian operator* on continuous spaces. The natural analogue on discrete spaces is $\mathbf{L}$, where we now treat $\mathbf{L}$ as a linear operator on vectors $\mathbf{u} \in \mathbb{R}^N$. This can be seen by noting that, if we take $\mathbf{W}$ to be the *unweighted* adjacency matrix, $\langle \mathbf{u}, \mathbf{L}\mathbf{u} \rangle = \mathbf{u}^\top \mathbf{L}\mathbf{u} = -\frac{1}{2} \sum_{(i,j) \in \mathcal{N}} (u_i - u_j)^2$ so, like its continuous counterpart, $\mathbf{L}$ measures the local smoothness of its domain. In fact, in this case $-\mathbf{L}$ is exactly the finite difference discretisation of $\nabla^2$ on a square grid in $N$-dimensional Euclidean space. This motivates the *discrete heat equation* on $G$,

$$\frac{d\mathbf{u}}{dt} = -\widetilde{\mathbf{L}}\mathbf{u}, \tag{3}$$

where we followed literature conventions by using the normalised variant whose spectrum is conveniently contained in $[0, 2]$ [Chung, 1997]. This has the solution $\mathbf{u}_t = \exp(-\widetilde{\mathbf{L}}t)\mathbf{u}_0$, where $\exp(-\widetilde{\mathbf{L}}t) = \lim_{n \to \infty} \left(1 - \frac{t\widetilde{\mathbf{L}}}{n}\right)^n$. The symmetric and positive semi-definite matrix

$$\mathbf{K}_{\text{diff}}(t) := \exp(-\widetilde{\mathbf{L}}t) \tag{4}$$

is referred to as the *heat kernel* or *diffusion kernel* [Smola and Kondor, 2003]. The exponentiation of the generator $\widetilde{\mathbf{L}}$, which by construction captures the *local* structure of $G$, leads to a kernel matrix $\mathbf{K}_{\text{diff}}$ which captures the graph's *global* structure. Upon discretisation of Eq. 3 with the backward Euler step (which is generally more stable than the forward), we have that

$$\mathbf{u}_{t+\delta t} = (\mathbf{I}_N + \delta t \widetilde{\mathbf{L}})^{-1} \mathbf{u}_t, \tag{5}$$

where the discrete time-evolution operator $\mathbf{K}_{\text{lap}}^{(1)} = (\mathbf{I}_N + \delta t \widetilde{\mathbf{L}})^{-1}$ is referred to as the 1-*regularised Laplacian kernel*. This is a member of the more general family of $d$-regularised Laplacian kernels,

$$\mathbf{K}_{\text{lap}}^{(d)} = (\mathbf{I}_N + \delta t \widetilde{\mathbf{L}})^{-d}, \tag{6}$$

for which we can construct an unbiased approximation using GRFs [Choromanski, 2023]. We predominantly consider $d = 2$ with the understanding that estimators for other values of $d$ (and the diffusion kernel) are straightforward to obtain – see App. 8.2. This demonstrates the intimate connection between the graph-diffusion and Laplacian kernels, and how a QMC scheme that improves the convergence of $\mathbf{K}_{\text{lap}}^{(2)}$ will be of broad interest.

## 2.2 Graph random features (GRFs)

Here we recall the GRF mechanism, which offers a rich playground for our novel antithetic termination QMC scheme. The reader should consult Choromanski [2023] (especially Algorithm 1.1) for a full discussion, but for convenience we provide a cursory summary.

Suppose we would like to estimate the matrix $(\mathbf{I}_N - \mathbf{U})^{-2}$, with $\mathbf{U} \in \mathbb{R}^{N \times N}$ a weighted adjacency matrix of a graph with $N$ nodes and no loops. Choromanski [2023] proposed a novel algorithm to construct a randomised estimator of this matrix. The author uses *graph random features* (GRFs) $\phi(i) \in \mathbb{R}^N$, with $i$ the index of one of the nodes, designed such that

$$(\mathbf{I}_N - \mathbf{U})_{ij}^{-2} = \mathbb{E}\left(\phi(i)^\top \phi(j)\right). \tag{7}$$

They construct $\phi(i)$ by taking $m$ random walks $\{\bar{\Omega}(k, i)\}_{k=1}^m$ on the graph out of node $i$, depositing a 'load' at every node that depends on i) the product of edge weights traversed by the subwalk and ii) the marginal probability of the subwalk. Importantly, each walk terminates with probability $p$ at every timestep. The $x$th component of the $i$th random feature is given by

$$\phi(i)_x \coloneqq \frac{1}{m} \sum_{k=1}^m \sum_{\omega_1 \in \Omega_{ix}} \frac{\widetilde{\omega}(\omega_1)}{p(\omega_1)} \mathbb{I}(\omega_1 \in \bar{\Omega}(k, i)), \tag{8}$$

where: $k$ enumerates $m$ random walks we sample out of node $i$; $\omega_1$ denotes a particular walk from the set of all walks $\Omega_{ix}$ between nodes $i$ and $x$; $\widetilde{\omega}(\omega_1)$ is the product of weights of the edges traversed by the walk $\omega_1$; $p(\omega_1)$ is the marginal probability that a walk contains a prefix subwalk $\omega_1$, given by $p(\omega_1) = ((1-p)/d)^{\text{len}(\omega_1)}$ in the simplest case of a walk of length $\text{len}(\omega_1)$ on a $d$-regular graph; $\mathbb{I}(\omega_1 \in \bar{\Omega}(k, i))$ is an indicator function that evaluates to 1 when the walk $\omega_1$ is a prefix subwalk of the $k$th random walk sampled from $i$ (itself denoted $\bar{\Omega}(k, i)$) and is 0 otherwise.

It is simple to see how Eq. 8 satisfies Eq. 7. By construction

$$\mathbb{E}\left[\mathbb{I}(\omega_1 \in \bar{\Omega}(k, i))\mathbb{I}(\omega_2 \in \bar{\Omega}(l, j))\right] = p(\omega_1)p(\omega_2) \tag{9}$$

for independent walks, whereupon

$$\mathbb{E}\left(\phi(i)^\top \phi(j)\right) = \sum_{x \in \mathcal{N}} \sum_{\omega_1 \in \Omega_{ix}} \sum_{\omega_2 \in \Omega_{jx}} \widetilde{\omega}(\omega_1)\widetilde{\omega}(\omega_2) = \sum_{\omega \in \Omega_{ij}} (\text{len}(\omega) + 1)\widetilde{\omega}(\omega) = (\mathbf{I} - \mathbf{U})_{ij}^{-2}. \tag{10}$$

This shows us that the estimator is unbiased. The central contribution of this work is a QMC scheme that induces correlations between the $m$ walks out of each node to suppress the variance of the estimator $\phi(i)^\top \phi(j)$ without breaking this unbiasedness.

# 3 q-GRFs and antithetic termination

We will now present our novel antithetic termination mechanism. It generalises the notion of antithetic variates – a common, computationally cheap variance-reduction technique when sampling in Euclidean space [Hammersley and Morton, 1956] – to the termination behaviour of random walks.

We have seen that, in the i.i.d. implementation of the GRF algorithm, each walker terminates independently with probability $p$ at every timestep. For a pair of i.i.d. walkers out of node $i$, this is implemented by independently sampling two *termination random variables* (TRVs) between 0 and 1 from a uniform distribution, $t_{1,2} \sim \text{Unif}(0, 1)$. Each walker terminates if its respective TRV is less than $p$, $t_{1,2} < p$. In contrast, we define the *antithetic* walker as follows.

**Definition 3.1** (Antithetic walkers). *We refer to a pair of walkers as* antithetic *if their TRVs are marginally distributed as* $t_{1,2} = \text{Unif}(0, 1)$ *but are offset by* $\frac{1}{2}$,

$$t_2 = \text{mod}_1\left(t_1 + \frac{1}{2}\right), \tag{11}$$

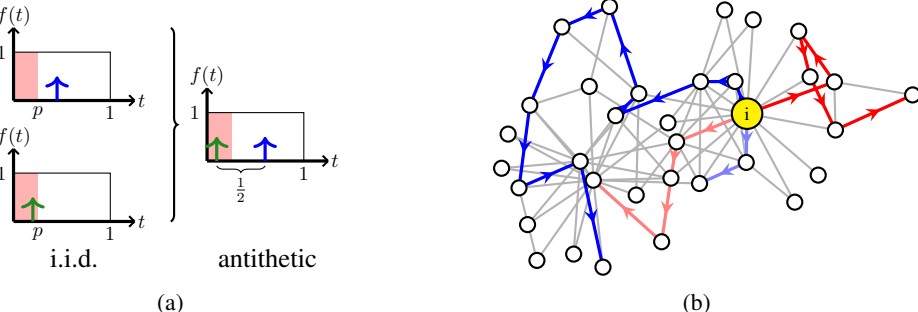

(a)  (b)

Figure 1: *Left*: schematic of the i.i.d. (GRF) and antithetic (q-GRF) mechanisms in termination space. $f(t)$ is the probability density of the termination random variable (TRV) $t$. Vertical arrows represent draws of $t$, with the walker terminating if they lie in the pink region where $t < p$. With q-GRFs the TRVs are offset by $\frac{1}{2}$, modifying the joint distribution over walk lengths. *Right*: demonstration with $4$ random walks on the `karate` graph, beginning at some node labelled $i$. The blue pair of antithetic walks (q-GRFs) have very different lengths; they cannot terminate simultaneously. The red pair of i.i.d. walks (GRFs) have similar lengths. We prove that q-GRFs give lower variance estimators of the 2-regularised Laplacian kernel.

*such that we have the conditional probability density*

$$p(t_2|t_1) = \delta\left(\mathrm{mod}_1(t_2 - t_1) - \frac{1}{2}\right). \tag{12}$$

In both schemes, the TRVs are resampled at every timestep until the corresponding walker terminates.

**Computational cost**: We note that the computational cost of generating an antithetic TRV according to Eq. 11 is no greater than the cost of generating an independent TRV, so this drop-in replacement in the GRF algorithm is cheap. We provide a schematic in Fig. 1a.

Since the marginal distributions over $t_i$ are unchanged our estimator remains unbiased, but the couplings between TRVs lead to statistical correlations between the walkers' terminations. Denoting by $s_1$ the event that walker 1 terminates at some timestep, $s_2$ the event that walker 2 terminates and $\bar{s}_{1,2}$ their complements, it is straightforward to convince oneself that for $p \leq \frac{1}{2}$

$$p(s_1) = p(s_2) = p, \quad p(\bar{s}_1) = p(\bar{s}_2) = 1 - p, \quad p(s_2|s_1) = 0,$$
$$p(\bar{s}_2|s_1) = 1, \quad p(s_2|\bar{s}_1) = \frac{p}{1-p}, \quad p(\bar{s}_2|\bar{s}_1) = \frac{1-2p}{1-p}. \tag{13}$$

This termination coupling modifies the joint probability distribution over walk lengths. In the i.i.d. scheme, the walks are independent and are of expected length

$$\mathbb{E}(\mathrm{len}(\omega)) = \frac{1-p}{p}. \tag{14}$$

These *marginal* expectations are preserved in the antithetic scheme, but now the expected length of one walk *conditioned on the length of the other* is

$$\mathbb{E}(\mathrm{len}(\omega_2)|\mathrm{len}(\omega_1) = m) = \frac{1-2p}{p} + 2\left(\frac{1-2p}{1-p}\right)^m, \tag{15}$$

which we derive in App. 8.3. It is straightforward to see that the two lengths are negatively correlated. Antithetic termination 'diversifies' the lengths of random walks we sample, preventing them from clustering together, and in the spirit of QMC this turns out to suppress the kernel estimator variance. See Fig. 1b for a schematic. We refer to random features constructed with antithetic walkers as *quasi-Monte Carlo graph random features* (q-GRFs).

Though this paper focuses on antithetic termination for GRFs, it is a more general QMC scheme applicable in algorithms that sample walks with geometrically distributed lengths. To illustrate this, we again direct the reader to App. 8.1 where we show how it can be used to improve numerical estimates of the PageRank vector [Page et al., 1998].

## 3.1 Theoretical results

In this section, we state and discuss our central theoretical results for the q-GRFs mechanism. Sec. 3.1.1 provides a sketch, but full proofs are deferred to App. 8.4. We remind the reader that results for $(\mathbf{I}_N - \mathbf{U})^{-2}$ are trivially applied to $\mathbf{K}_{\text{lap}}^{(2)}$ (see App. 8.2).

**Theorem 3.2** (Antithetic termination is better than i.i.d.). *For any graph, q-GRFs will give lower variance on estimators of $(\mathbf{I}_N - \mathbf{U})^{-2}$ than regular GRFs provided either i) the termination probability $p$ is sufficiently small or ii) the spectral radius $\rho(\mathbf{U})$ is sufficiently small.*

By 'sufficiently small' we mean that for a fixed $\rho(\mathbf{U})$ there exists some value of $p$ below which antithetic termination will outperform i.i.d.. Likewise, for fixed $p$ there exists some value of $\rho(\mathbf{U})$. These conditions turn out to not be too restrictive in our experiments; antithetic termination is actually very effective at $p = 0.5$ which we use for practical applications.

Considering Eq. 11 carefully, it is easy to see that the termination probabilities in Eq. 13 are not particular to a TRV offset equal to $\frac{1}{2}$, and in fact hold for any offset $\Delta$ satisfying $p \leq \Delta \leq 1 - p$. An immediate corollary is as follows.

**Corollary 3.3** (Maximum size of an antithetic ensemble). *For a termination probability $p$, up to $\lfloor p^{-1} \rfloor$ random walkers can all be conditioned to exhibit mutually antithetic termination.*

This is achieved by offsetting their respective TRVs by $\Delta = p$. The resulting *antithetic ensemble* will have lower kernel estimator variance than the equivalent number of i.i.d. walkers or a set of mutually independent antithetic pairs. We make one further interesting remark.

**Theorem 3.4** (Termination correlations beyond antithetic). *A pair of random walkers with TRVs offset by $p(1 - p) < \Delta < p$ will exhibit lower variance on estimators of $(\mathbf{I}_N - \mathbf{U})^{-2}$ than independent walkers, provided either i) the termination probability $p$ is sufficiently small or ii) the spectral radius of the weighted adjacency matrix $\rho(\mathbf{U})$ is sufficiently small.*

This provides an upper limit of $\lfloor (p(1 - p))^{-1} \rfloor$ on the number of walkers we can simultaneously correlate before we can no longer guarantee that coupling in a further walker's TRV will be better than sampling it independently. Intuitively, Theorem 3.4 tells us that we can space TRVs even more closely than $p$, allowing us to increase the number of simultaneously anticorrelated random walkers at the cost of the strength of negative correlations between walkers with neighbouring TRVs.

### 3.1.1 Proof sketch

In this section, we will outline a proof strategy for the results reported earlier in this section. Full technical details are reported in App. 8.4.

From its Taylor expansion, the $(ij)$-th element of $(\mathbf{I}_N - \mathbf{U})^{-2}$ is nothing other than a sum over all possible walks between the nodes $i$ and $j$, weighted by their lengths and the respective products of edge weights. GRFs use a Monte Carlo scheme to approximate this sum by sampling such walks at random – concretely, by first sampling separate random walks out of nodes $i$ and $j$ and summing contributions wherever they intersect. In order to sample the space of walks between nodes $i$ and $j$ more efficiently, it follows that we should make our ensemble of walks out of each node more diverse. q-GRFs achieve this by inducing negative correlations such that they are different lengths.

In more detail, it is clear that the variance of the kernel estimator will depend upon the expectation of the square of $\phi(i)^\top \phi(j)$. Each term in the resulting sum will take the form

$$\sum_{x,y \in \mathcal{N}} \sum_{\omega_1 \in \Omega_{ix}} \sum_{\omega_2 \in \Omega_{jx}} \sum_{\omega_3 \in \Omega_{iy}} \sum_{\omega_4 \in \Omega_{jy}} \frac{\widetilde{\omega}(\omega_1)}{p(\omega_1)} \frac{\widetilde{\omega}(\omega_2)}{p(\omega_2)} \frac{\widetilde{\omega}(\omega_3)}{p(\omega_3)} \frac{\widetilde{\omega}(\omega_4)}{p(\omega_4)} \tag{16}$$
$$\cdot p(\omega_1 \in \bar{\Omega}(k_1, i), \omega_3 \in \bar{\Omega}(k_2, i)) p(\omega_2 \in \bar{\Omega}(l_1, j), \omega_4 \in \bar{\Omega}(l_2, j)),$$

where we direct the reader to Sec. 2.2 for the symbol definitions. Supposing that all edge weights are equal (an assumption we relax later), the summand is a function of the *length* of each of the walks $\omega_{1,2,3,4}$. It is then natural to write the sum over all walks between nodes $i$ and $x$ as a sum over walk lengths $m$, with each term weighted by a combinatorial factor that counts the number of walks of said length. This factor is $(\mathbf{A}^m)_{ix}$, with $\mathbf{A}$ the unweighted adjacency matrix. That is,

$$\sum_{\omega_1 \in \Omega_{ix}} (\cdot) = \sum_{m=0}^{\infty} (\mathbf{A}^m)_{ix} (\cdot). \tag{17}$$

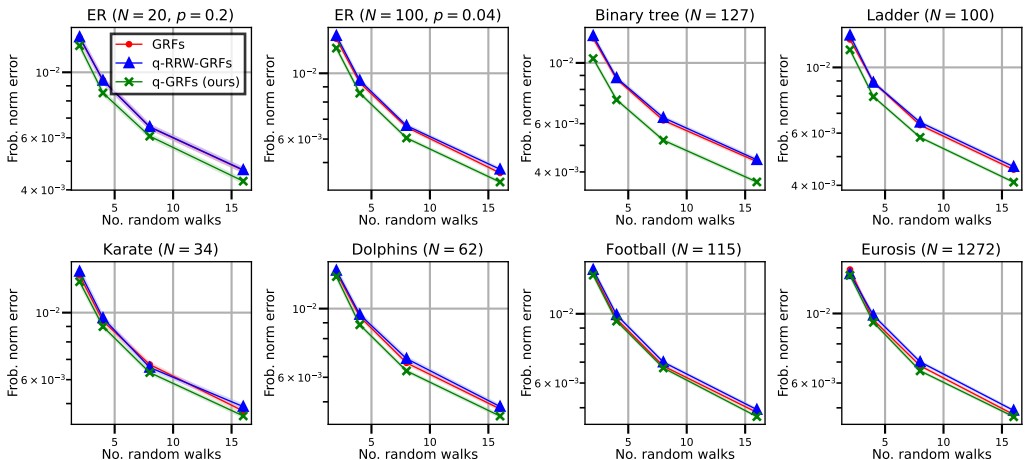

Figure 2: Relative Frobenius norm error of estimator of the 2-regularised Laplacian kernel with GRFs (red circle) and q-GRFs (green cross). Lower is better. We also include q-RRW-GRFs [Choromanski, 2023] (blue triangle), instantiated with $f = \exp$, as a benchmark. Our novel q-GRFs perform the best on every graph considered. $N$ is the number of nodes and, for the Erdős-Rényi (ER) graphs, $p$ is the edge-generation probability. One standard deviation is shaded but it is too small to easily see.

$(\mathbf{A}^m)_{ix}$ is readily written as its eigendecomposition, $\sum_{p=1}^N \lambda_p^m k_{pi} k_{px}$, with $\lambda_p$ the $p$th eigenvalue and $k_{pi}$ the $i$-th coordinate of the $p$-th eigenvector $\boldsymbol{k}_p$. We put these into Eq. 16 and perform the sums over each walk length $m_{1,2,3,4}$ from 1 to $\infty$, arriving at

$$\sum_{x,y \in \mathcal{N}} \sum_{k_1, k_2, k_3, k_4} f(\lambda_1, \lambda_2, \lambda_3, \lambda_4) k_{1i} k_{1x} k_{2j} k_{2x} k_{3i} k_{3y} k_{4j} k_{4y} \tag{18}$$

where $f(\lambda_1, \lambda_2, \lambda_3, \lambda_4)$ is a function of the eigenvalues of $\mathbf{A}$ that depends on whether we correlate the terminations of the walkers. Since the eigenvectors are orthogonal we have that $\sum_{x \in \mathcal{N}} k_{1x} k_{2x} = \delta_{12}$, so this reduces to

$$\sum_{k_1, k_3} f(\lambda_1, \lambda_1, \lambda_3, \lambda_3) k_{1i} k_{1j} k_{3i} k_{4j}. \tag{19}$$

Our task becomes to prove that this expression becomes smaller when we induce antithetic termination. We achieve this by showing that a particular matrix (with elements given by the difference in $f$ between the schemes) is negative definite: a task we achieve by appealing to Weyl's perturbation inequality [Bai et al., 2000].

## 4 Experiments

In this section we report on empirical evaluations of q-GRFs. We confirm that they give lower kernel estimator variance than regular GRFs and show that this often leads to substantially better performance in downstream tasks, including simulation of graph diffusion, $k$-means node clustering and kernel regression for node attribute prediction. We use ensembles of antithetic pairs (Def. 3.1).

### 4.1 Estimation of the 2-regularised Laplacian kernel

We begin with the simplest of tasks: estimation of the 2-regularised Laplacian kernel,

$$\mathbf{K}_{\text{lap}}^{(2)} = (\mathbf{I}_N + \sigma^2 \widetilde{\mathbf{L}})^{-2}, \tag{20}$$

where $\widetilde{\mathbf{L}} \in \mathbb{R}^{N \times N}$ is the symmetrically normalised Laplacian. $0 < \sigma < 1$ is a regulariser. We use both GRFs and q-GRFs to generate unbiased estimates $\widetilde{\mathbf{K}}_{\text{lap}}^{(2)}$ (see App. 8.2), then compute the relative Frobenius norm $\|\mathbf{K}_{\text{lap}}^{(2)} - \widetilde{\mathbf{K}}_{\text{lap}}^{(2)}\|_{\text{F}}^2 / \|\mathbf{K}_{\text{lap}}^{(2)}\|_{\text{F}}^2$ between the true and approximated kernel matrices. This enables us to compare the quality of the estimators. As a benchmark, we also include

an implementation of the high-level reinforced random walk QMC mechanism suggested (but not tested) by Choromanski [2023]. We choose the exponential mapping as the reinforcement function $f$ (used to downweight the probability of traversing previously-visited edges), although the optimal choice remains an open problem. We refer to this mechanism as *q-RRW-GRFs* to disambiguate from our instantiation of q-GRFs (which uses antithetic termination).

Fig. 2 presents the results for a broad class of graphs: small Erdős-Rényi, larger Erdős-Rényi, a binary rooted tree, a ladder, and four real-world examples from Ivashkin [2023] (`karate`, `dolphins`, `football` and `eurosis`). We consider 2, 4, 8 and 16 walks, taking 100 repeats for the variance of the approximation error. We use the regulariser $\sigma = 0.1$ and the termination probability $p = 0.5$.

The quality of kernel approximation naturally improves with the number of walkers. Inducing antithetic termination consistently reduces estimator variance, with our q-GRF mechanism outperforming regular GRFs in every case. The exact size of the gain depends on the particular graph (according to the closed forms derived in App. 8.4), but improvement is always present. It is intriguing that tree-like and planar graphs tend to enjoy a bigger gap; we defer a rigorous theoretical analysis to future work. Meanwhile, the q-RRW-GRF variant with exponential $f$ is often worse than the regular mechanism and is substantially more expensive. We do not include it in later experiments.

## 4.2 Scalable and accurate simulation of graph diffusion

In Sec. 2.1, we noted that the Laplacian $\mathbf{L}$ (or $\widetilde{\mathbf{L}}$) is the natural operator to describe diffusion on discrete spaces and that the 1-regularised Laplacian kernel constitutes the corresponding discrete time-evolution operator. Here we will show how, by leveraging q-GRFs to provide a lower-variance decomposition of $\mathbf{K}_{\text{lap}}^{(2)}$, we can simulate graph diffusion in a scalable and accurate way.

Choosing a finite (even) number of discretisation timesteps $N_t$, we can approximate the final state by

$$\mathbf{u}_t \simeq \widetilde{\mathbf{u}}_t := \left[ (\mathbf{I}_N + \frac{t}{N_t}\widetilde{\mathbf{L}})^{-2} \right]^{\frac{N_t}{2}} \mathbf{u}_0 = \mathbf{K}_{\text{lap}}^{(2)\frac{N_t}{2}} \mathbf{u}_0 \tag{21}$$

This is equivalent to approximating the action of the diffusion kernel $\mathbf{K}_{\text{diff}}$ (see App. 8.2). We can efficiently compute this using our GRF or q-GRF decomposition of $\mathbf{K}_{\text{lap}}^{(2)}$ and compare the accuracy of reconstruction of $\mathbf{u}_t$.[2] In particular, we take an initial one-hot state $\mathbf{u}_0 = (1, 0, 0, ..., 0)^\top$ and simulate diffusion for $t = 1$ divided into $N_t = 1000$ timesteps, using $m = 10$ walkers and a termination probability $p = 0.5$. Fig. 3 gives a schematic. We average the MSE of $\widetilde{\mathbf{u}}_t$ over 1000 trials. Table 1 reports the results; q-GRFs approximate the time evolution operator more accurately so consistently give a lower simulation error. Since the time-evolution operator is applied repeatedly, even modest improvements in its approximation can lead to a substantially better reconstruction of the final state. In extreme cases the simulation error is halved.

Table 1: MSE of the approximation of the final state $\mathbf{u}_t$ by repeated application of an approximation of the discrete time-evolution operator, constructed with (q-)GRFs. Brackets give one standard deviation.

| Graph | $N$ | Sim error, $\text{MSE}(\widetilde{\mathbf{u}}_t)$ GRFs | q-GRFs |
|---|---|---|---|
| Small ER | 20 | 0.0210(5) | **0.0160(3)** |
| Larger ER | 100 | 0.0179(9) | **0.0085(3)** |
| Binary tree | 127 | 0.0161(6) | **0.0106(3)** |
| Ladder | 100 | 0.0190(8) | **0.0105(3)** |
| karate | 34 | 0.066(2) | **0.054(1)** |
| dolphins | 62 | 0.0165(4) | **0.0139(3)** |
| football | 115 | 0.0170(3) | **0.0160(2)** |
| eurosis | 1272 | 0.089(1) | **0.084(1)** |

Figure 3: Schematic of diffusion on a small graph, with heat initially localised on a single node spreading under the action of the Laplace operator. We use (q-)GRFs to estimate the quantity on the right.

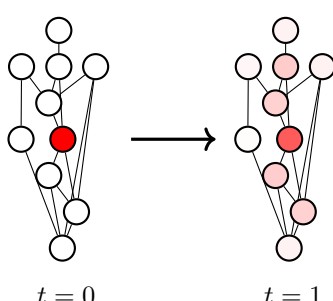

$t = 0$        $t = 1$

---

[2]To be asymptotically unbiased, we should construct $N_t/2$ independent estimates of $\mathbf{K}_{\text{lap}}^{(2)}$ (see App. 8.2). However, here we observe that the error is small even when we apply the same approximation repeatedly, i.e. estimate $\mathbf{K}_{\text{lap}}^{(2)}$ once and use it to simulate diffusion at every timestep, despite this remaining biased as $N_t \to \infty$.

### 4.3 Kernelised $k$-means clustering for graph nodes

Next we evaluate the performance of GRFs and q-GRFs on the task of assigning nodes to clusters using the graph kernel, following the approach described by Dhillon et al. [2004]. We first run the algorithm to obtain $N_c = 2$ clusters with the exact 2-regularised Laplacian kernel, then compare the results when we use its approximation via GRFs and q-GRFs. In each case, we report the *clustering error*, defined by

$$E_c := \frac{\text{no. wrong pairs}}{N(N-1)/2}. \qquad (22)$$

This is simply the number of misclassified pairs (in the sense of being assigned to the same cluster when the converse is true or vice versa) divided by the total number of pairs. The results are less straightforward because curiously the clustering error does not generally vary monotonically with the variance on the kernel estimator, but nonetheless in six out of eight cases q-GRFs provide equally good or better results.

Table 2: Errors in kernelised $k$-means clustering when approximating the Gram matrix with q-GRFs and GRFs. Lower is better.

| Graph | $N$ | Clustering error, $E_c$ | |
| --- | --- | --- | --- |
| | | GRFs | q-GRFs |
| karate | 34 | 0.11 | **0.05** |
| dolphins | 62 | 0.40 | **0.38** |
| polbooks | 105 | **0.28** | **0.28** |
| football | 115 | 0.10 | **0.09** |
| eu-core | 1005 | **0.04** | 0.05 |
| databases | 1046 | 0.17 | **0.11** |
| eurosis | 1285 | **0.15** | 0.19 |
| citeseer | 3300 | 0.02 | **0.01** |

### 4.4 Kernel regression for node attribute prediction

Lastly, we consider the problem of kernel regression on a triangular mesh graph. Each node is associated with a normal vector $\boldsymbol{v}^{(i)}$, equal to the mean of the normal vectors of its surrounding faces. The task is to predict the directions of missing vectors (a random 5% split) from the remainder. We consider five meshes of different sizes, available from Dawson-Haggerty [2023]. Our prediction for the direction of the normal vector corresponding to the $i$th node is

$$\widetilde{\boldsymbol{v}}^{(i)} := \frac{\sum_j \mathbf{K}_{\text{lap}}^{(2)}(i,j)\boldsymbol{v}^{(j)}}{\sum_j \mathbf{K}_{\text{lap}}^{(2)}(i,j)}, \qquad (23)$$

Table 3: $1 - \cos\theta$ between true and predicted node vectors when approximating the Gram matrix with q-GRFs and GRFs. Lower is better. Brackets give one standard deviation.

| Graph | $N$ | Pred error, $1 - \cos\theta$ | |
| --- | --- | --- | --- |
| | | GRFs | q-GRFs |
| cylinder | 210 | 0.104(1) | **0.101(1)** |
| teapot | 480 | 0.0531(5) | **0.0493(5)** |
| idler-riser | 782 | 0.0881(6) | **0.0852(5)** |
| busted | 1941 | 0.00690(4) | **0.00661(4)** |
| torus | 4350 | 0.00131(1) | **0.00120**(1) |

where $j$ sums over the vertices with known vectors (a 95% split). This is simply a linear combination of all the training node normal vectors, weighted by their respective kernel evaluations. We compute the average angular error $1 - \cos\theta$ between the prediction $\widetilde{\boldsymbol{v}}^{(i)}$ and groundtruth $\boldsymbol{v}^{(i)}$ across the missing vectors, comparing the result when $\mathbf{K}_{\text{lap}}^{(2)}$ is approximated with GRFs and q-GRFs with $m = 6$ random walks at a termination probability $p = 0.5$. The regulariser is $\sigma = 0.1$. q-GRFs enjoy lower kernel estimator variance so consistently give better predictions for the missing vectors.

## 5 Conclusion

We have proposed a novel class of quasi-Monte Carlo graph random features (q-GRFs) for unbiased and efficient estimation of kernels defined on the nodes of a graph. We have proved that our new algorithm, which induces negative statistical correlations between the lengths of graph random walks via antithetic termination, enjoys better convergence properties than its regular predecessor (GRFs). This very often permits better performance in empirical tasks and for some applications the improvement is substantial. Our work ushers in further research in this new domain of quasi-Monte Carlo methods for kernels on combinatorial objects. It may be of broader interest including for algorithms that sample random walks.

# 6    Broader impacts and limitations

We envisage several possible **impacts** of our novel algorithm. First, graphs provide a natural way to describe systems characterised by complex biological interactions such as the proteins at the proteome scale or drugs in the body [Ingraham et al., 2019]. Our novel QMC algorithm might be of translational impact in this **bioinformatics** setting. Antithetic termination is at its heart a procedure to improve the sampling efficiency of random walks, so it could also help mitigate the notoriously high **energy and carbon** cost of large models [Strubell et al., 2019]. Lastly, we believe our results are of **intrinsic interest** as the first (to our knowledge) rigorously studied QMC scheme defined on a combinatorial object. They might spur further research in this new domain. Our work is **foundational** with no immediate direct negative societal impacts that we can see. However, it is important to note that increases in scalability afforded by GRF and q-GRF algorithms could amplify risks of graph-based machine learning, either from bad actors or as unintended consequences.

The work also has some **limitations** and **directions for future research**. First, although we have derived closed-form expressions for the kernel estimator variance with GRFs and q-GRFs in App. 8.4, they are still complicated functions of the spectra of the respective graph Laplacians. Understanding **what characterises graphs that particularly benefit** from antithetic termination (empirically, tree-like and planar graphs) is an important future direction. Moreover, our scheme only correlates walk lengths. A more sophisticated mechanism that **couples walk directions** might do even better. Lastly, further work is needed to fully understand the applicability of antithetic termination **beyond the GRF setting**.

# 7    Relative contributions and acknowledgements

IR devised the antithetic termination scheme, proved all theoretical results and ran the experiments in Secs 4.1, 4.2 and 4.4. KC provided crucial support throughout, particularly: running the clustering experiment in Sec. 4.3, showing how $d$-regularised Laplacian kernels can be used to approximate the graph diffusion process, and proposing to apply q-GRFs to the problems in Secs 4.2 and 4.4. AW gave important guidance and feedback on the manuscript.

IR acknowledges support from a Trinity College External Studentship. AW acknowledges support from a Turing AI fellowship under grant EP/V025279/1 and the Leverhulme Trust via CFI.

We thank Austin Tripp and Kenza Tazi for their thoughtful feedback on earlier versions of the text, and Michael Ren for his excellent suggestion to treat the negative definite property perturbatively. We are also grateful to the anonymous reviewers.

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

# 8 Appendices

**Notation**: to reduce the burden of summation indices, in places we use Dirac's bra-ket notation from quantum mechanics. $|y\rangle$ can be interpreted as the vector $\boldsymbol{y}$ and $\langle y|$ as $\boldsymbol{y}^\top$.

## 8.1 Antithetic termination for PageRank

Here we demonstrate the broader applicability of antithetic termination beyond the GRF setting by showing how it can be used to improve the convergence of numerical estimators of the PageRank vector. The treatment is similar to App. 8.4; the reader may benefit from reviewing this section first or in parallel.

The PageRank vector is a measure of the relative importance of the nodes in a graph, proposed by Page et al. [1998] to rank websites in search engine results. It is the steady state of a Markov chain with a $N \times N$ transition matrix

$$\mathbf{P} := (1-p)\mathbf{T} + \frac{p}{N}\mathbf{E}, \tag{24}$$

where $p \in (0,1)$ is a scalar and $N = |\mathcal{N}|$ is the number of graph nodes. The matrix $\mathbf{E} := [1]_{i,j\in\mathcal{N}}$ has entries that are all equal to 1, and the matrix $\mathbf{T}$ is given by

$$\mathbf{T}_{ij} = \begin{cases} \frac{1}{d_i} & \text{if } (i,j) \in \mathcal{E} \\ 0 & \text{otherwise} \end{cases} \tag{25}$$

with $d_i = \sum_{j\in\mathcal{N}} \mathbb{I}((i,j) \in \mathcal{E})$, the degree (number of neighbours) of the $i$th node. The state space of the Markov chain is the nodes $\mathcal{N}$. $\mathbf{T}$ encodes the behaviour of a simple random walk.

Formally, we are solving for the vector $\boldsymbol{\pi} \in \mathbb{R}^N$ such that

$$\boldsymbol{\pi}^\top \mathbf{P} = \boldsymbol{\pi}^\top, \quad \boldsymbol{\pi}^\top \mathbf{1} = 1 \tag{26}$$

where $\mathbf{1} := (1,1,...,1)$. Physically, $\boldsymbol{\pi}_j$ is the stationary probability that a surfer is located at node $j$, supposing that at every timestep it either i) moves to one of its neighbouring nodes (with probability $1-p$) or ii) teleports to a random node (with probability $p$).

For large graphs, $\boldsymbol{\pi}$ becomes very expensive to compute exactly, motivating a host of research dedicated to approximating it. Solving Eq. 26 and Taylor expanding $(1-(1-p)\mathbf{T})^{-1}$, the solution is given by

$$\boldsymbol{\pi}_i = \frac{p}{N} \sum_{j\in\mathcal{N}} \sum_{k=0}^\infty (1-p)^k \mathbf{T}_{ji}^k. \tag{27}$$

Note that the term

$$p\sum_{k=0}^\infty (1-p)^k \mathbf{T}_{ji}^k \tag{28}$$

is nothing other than a sum over all walks between nodes $j$ and $i$, weighted by $p(1-p)^k \prod_{i=1}^k \frac{1}{d_i}$ with $k$ the walk length and $d_i$ the degree of the $i$th node visited on the walk. Supposing a walker terminates with probability $p$ at each timestep and picks a neighbour to visit at random, this weight is equal to the probability of sampling a particular walk between nodes $j$ and $i$, allowing us to write

$$\boldsymbol{\pi}_i = \frac{1}{N} \sum_{j\in\mathcal{N}} p\,(\text{walker starting at } j \text{ terminates at } i)\,. \tag{29}$$

This invites the Monte Carlo estimate

$$\widehat{\boldsymbol{\pi}}_i = \frac{1}{Nm} \sum_{j\in\mathcal{N}} \sum_{l=1}^m \mathbb{I}(\text{walker starting at } j \text{ terminates at } i) \tag{30}$$

with $m$ the number of walks sampled out of each node. We have arrived at the algorithm proposed by Fogaras et al. [2005]: simulate $m$ walks out of each node $j$ of the graph, and compute the estimator $\widehat{\boldsymbol{\pi}}_i$ as the proportion of walkers terminating at node $i$.

Since we are sampling random walks that terminate with probability $p$ at every timestep, this is a natural setting to try antithetic termination. The following is true.

**Theorem 8.1** (Superiority of antithetic termination for PageRank estimation). *Supposing an ensemble of random walkers exhibits either i) i.i.d. or ii) antithetic termination and $p \leq \frac{1}{2}$,*

$$Var_{antithetic}(\widehat{\boldsymbol{\pi}}_i) \leq Var_{i.i.d.}(\widehat{\boldsymbol{\pi}}_i). \tag{31}$$

**Proof**: It is sufficient to consider the behaviour of a single pair of walkers out of node $j$ that are either i.i.d. or exhibit antithetic termination. Since in both cases the marginal probability of walkers terminating at node $i$ is identical, the estimators are unbiased. It is sufficient to determine whether the expectation value

$$\mathbb{E}\left(\mathbb{I}[\Omega_1 \text{ terminates at node } i\,]\,\mathbb{I}[\Omega_2 \text{ terminates at node } i\,]\right) = p(\text{both terminate at } i) \tag{32}$$

is suppressed by antithetic termination, where $\Omega_{1,2}$ denotes two walks simulated out of $j$.

If the two walkers are independent, Eq. 32 evaluates to

$$p(\text{both terminate at } i) = p^2 \left(\sum_{\alpha=1}^{N} \frac{1}{1 - (1-p)\lambda_\alpha} \langle j|k_\alpha\rangle \langle k_\alpha|i\rangle\right)^2 \tag{33}$$

where $\lambda_\alpha$ and $|k_\alpha\rangle$ denote the $\alpha$-th eigenvalue and eigenvector of the transition matrix $\mathbf{T}$ respectively. Meanwhile, for walkers with antithetic termination, it is straightforward to derive that

$$p(\text{len}(\Omega_2) = l_2 | \text{len}(\Omega_1) = l_1) = \begin{cases} \left(\frac{1-2p}{1-p}\right)^{l_2} \frac{p}{1-p} & \text{if } l_2 < l_1, \\ 0 & \text{if } l_2 = l_1, \\ \left(\frac{1-2p}{1-p}\right)^{l_1} (1-p)^{l_2-l_1-1}p & \text{if } l_2 > l_1, \end{cases} \tag{34}$$

where the walk lengths are no longer independent. It follows that

$$p(\Omega_2 \text{ terminates at } i \mid \text{len}(\Omega_1) = l_1) = \sum_{l_2=0}^{l_1-1} \left(\frac{1-2p}{1-p}\right)^{l_2} \frac{p}{1-p} \mathbf{T}_{ji}^{l_2} +$$
$$\sum_{l_2=l_1+1}^{\infty} \left(\frac{1-2p}{1-p}\right)^{l_1} (1-p)^{l_2-l_1-1}p\mathbf{T}_{ji}^{l_2}. \tag{35}$$

We then sum over all the possible walks $\Omega_1$ at each length $l_1$. After tedious but straightforward algebra,

$$p(\text{both terminate at } i) = \frac{p^2}{1-p} \sum_{\alpha,\beta=1}^{N} \langle j|k_\alpha\rangle \langle k_\alpha|i\rangle \langle j|k_\beta\rangle \langle k_\beta|i\rangle \left(\frac{(1-p)\lambda_\alpha}{1-(1-p)\lambda_\alpha}\right.$$
$$\left. + \frac{(1-p)\lambda_\beta}{1-(1-p)\lambda_\beta}\right) \frac{1}{1-(1-2p)\lambda_\alpha\lambda_\beta}. \tag{36}$$

To assess whether antithetic termination suppresses the variance, we need to consider the *difference* between Eqs 33 and 36. This evaluates to

$$\Delta p(\text{both terminate at } i) = \sum_{\alpha,\beta=1}^{N} -p^2 \frac{1}{1-(1-p)\lambda_\alpha} \frac{1}{1-(1-p)\lambda_\beta} \frac{(\lambda_\alpha-1)(\lambda_\beta-1)}{1-(1-2p)\lambda_\alpha\lambda_\beta} \tag{37}$$
$$\cdot \langle j|k_\alpha\rangle \langle k_\alpha|i\rangle \langle j|k_\beta\rangle \langle k_\beta|i\rangle.$$

For any pair of nodes $i, j$, the products $\langle j|k_\alpha\rangle \langle k_\alpha|i\rangle$ and $\langle j|k_\beta\rangle \langle k_\beta|i\rangle$ can be treated as the $\alpha$- and $\beta$- coordinates of an $N$-dimensional vector. Then we simply require that the matrix whose $\alpha, \beta$-th element is given by

$$-p^2 \frac{1}{1-(1-p)\lambda_\alpha} \frac{1}{1-(1-p)\lambda_\beta} \frac{(\lambda_\alpha-1)(\lambda_\beta-1)}{1-(1-2p)\lambda_\alpha\lambda_\beta} \tag{38}$$

is negative definite, which is manifestly the case provided $p \leq \frac{1}{2}$ (e.g. by Taylor expanding the denominator and observing that it is a sum of terms symmetric in $\lambda_{\alpha,\beta}$ with negative coefficients). It follows that $\Delta p(\text{both terminate at } i) \leq 0$, and the claim in Eq. 31 follows. $\square$

We run a simple numerical test to check Thm 8.1 for a number of graphs (see Sec. 4.1 of the main text for details), and find that using antithetic termination does indeed improve the quality of estimation of the PageRank vector. Fig. 4 reports the results, showing the error on the estimator $\|\boldsymbol{\pi} - \widehat{\boldsymbol{\pi}}\|_2^2$ for both schemes with 2, 4, 8 and 16 walkers. We choose a termination probability $p = 0.3$. As per the theoretical guarantees, antithetic random walks consistently yield more accurate estimators $\widehat{\boldsymbol{\pi}}$.

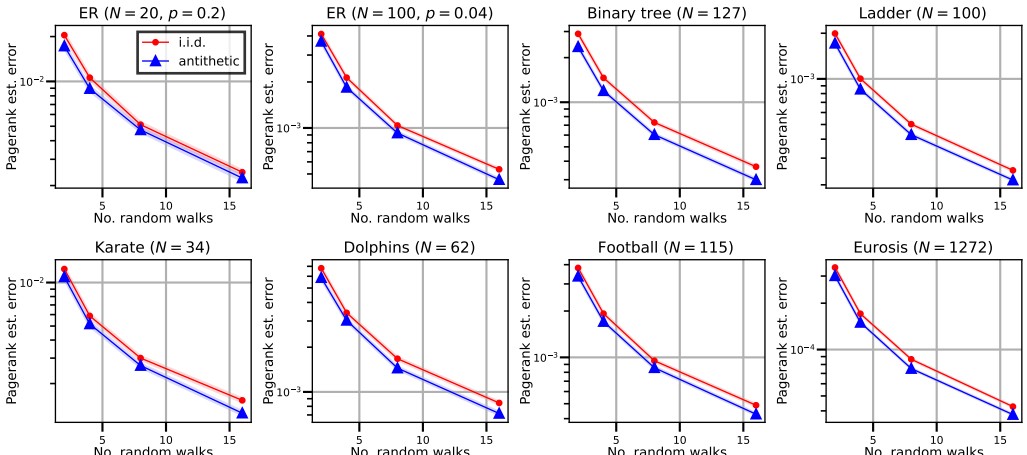

Figure 4: Error of the estimator of the PageRank vector $\|\boldsymbol{\pi} - \widehat{\boldsymbol{\pi}}\|_2^2$ when approximated using i) i.i.d. and ii) antithetic random walks. Lower is better. Antithetic termination yields a more accurate estimate for every graph considered. $N$ is the number of nodes and, for the Erdős-Rényi (ER) graphs, $p$ is the edge-generation probability. One standard deviation is shaded but it is too small to easily see.

## 8.2 On the approximation of the $d$-regularised Laplacian using GRFs

In this appendix, we demonstrate how to approximate the $d$-regularised Laplacian $\mathbf{K}_{\text{lap}}^{(d)}$ with GRFs.

Recall that GRFs provide an estimator to the quantity $(\mathbf{I}_N - \mathbf{U})^{-2}$ where $\mathbf{U}$ is a weighted adjacency matrix. Recall also that the matrix elements of the symmetrically normalised Laplacian $\widetilde{\mathbf{L}}$ are given by

$$\widetilde{\mathbf{L}}_{ij} = \begin{cases} 1 & \text{if } i = j, \\ -\dfrac{\mathbf{W_{ij}}}{\sqrt{\deg_{\mathbf{W}}(i)\deg_{\mathbf{W}}(j)}} & \text{if } i \in \mathcal{N}(j). \end{cases} \tag{39}$$

where $\deg_{\mathbf{W}}(i) = \sum_{j \in \mathcal{N}} \mathbf{W}_{ij}$ is the weighted degree of the node $i$. We are often interested in situations where $\mathbf{W} = \mathbf{A}$, an unweighted adjacency matrix. Now note that

$$\mathbf{K}_{\text{lap } ij}^{(2)} = (\mathbf{I}_N + \sigma^2\widetilde{\mathbf{L}})_{ij}^{-2} = (1 + \sigma^2)^{-2} (\mathbf{I}_N - \mathbf{U})_{ij}^{-2} \tag{40}$$

where we defined the matrix $\mathbf{U}$ with matrix elements

$$\mathbf{U}_{ij} = \frac{\sigma^2}{1 + \sigma^2} \frac{\mathbf{W}_{ij}}{\sqrt{\deg_{\mathbf{W}}(i)\deg_{\mathbf{W}}(j)}}. \tag{41}$$

This is itself a weighted adjacency matrix. It follows that, by estimating $(\mathbf{I}_N - \mathbf{U})^{-2}$ with GRFs, we can estimate $\mathbf{K}_{\text{lap}}^{(2)}$. This was reported by Choromanski [2023].

Supposing that we have constructed a GRF estimator

$$\mathbf{K}_{\text{lap}}^{(2)} = \mathbb{E}\left[\mathbf{C}\mathbf{C}^\top\right] \tag{42}$$

where the matrix $\mathbf{C} \in \mathbb{R}^{N \times N}$ has rows $\mathbf{C}_i := \frac{1}{1+\sigma^2}\phi(i)^\top$,[3] we note that it is straightforward to construct the 1-regularised Laplacian kernel estimator

$$\mathbf{K}_{\text{lap}}^{(1)} = \mathbb{E}\left[\mathbf{C}\mathbf{D}^\top\right] \tag{43}$$

by taking $\mathbf{D} := \left(\mathbf{I}_N + \sigma^2\widetilde{\mathbf{L}}\right)^\top \mathbf{C}$. By multiplying independent copies of the estimators $\widehat{\mathbf{K}}_{\text{lap}}^{(2)}$ and $\widehat{\mathbf{K}}_{\text{lap}}^{(1)}$, it is then trivial to obtain an unbiased estimator $\mathbf{K}_{\text{lap}}^{(d)}$ for arbitrary $d \in \mathbb{N}$. For instance,

$$\widehat{\mathbf{K}}_{\text{lap}}^{(5)} = \mathbf{C}_1\mathbf{C}_1^\top\mathbf{C}_2\mathbf{C}_2^\top\mathbf{C}_3\mathbf{D}_3^\top. \tag{44}$$

---

[3]Technically, for unbiasedness the random walks used for $\mathbf{C}$ and $\mathbf{C}^\top$ should be different, though this rarely makes any observable difference. See App. 8.7 for further discussion.

Note also that

$$\mathbf{K}_{\text{diff}} \coloneqq \exp(-\sigma^2 \widetilde{\mathbf{L}} t) = \lim_{n \to \infty} \left(1 - \frac{\sigma^2 \widetilde{\mathbf{L}} t}{n}\right)^n \simeq \left(1 - \frac{\sigma^2 \widetilde{\mathbf{L}} t}{d}\right)^d \tag{45}$$

with $d \gg 1$ the number of discretisation timesteps. We again included a regulariser $\sigma$. This shows that the ability to approximate the $d$-regularised Laplacian kernel for arbitrary $d$ permits us approximate the diffusion kernel, with the corresponding estimator asymptotically unbiased as $d \to \infty$. In Sec. 4.2 we see that the quality of approximation (in this case measured by the accuracy of simulated diffusion) is good even with modest $d$.

We also note that the variance of estimates $\widehat{\mathbf{K}}_{\text{lap}}^{(d)}$ and $\widehat{\mathbf{K}}_{\text{diff}}$ constructed in this manner will vary monotonically with the variance of the approximation of the constituent estimators $\widehat{\mathbf{K}}_{\text{lap}}^{(2)}$, so improvements provided by our QMC scheme (and any theoretical guarantees) will follow for this more general class of kernels.

### 8.3   Walk lengths with antithetic termination are anticorrelated (derivation of Eq. 15)

In this appendix we derive Eq. 15, which gives the expected length of some walk $\omega_2$ given that its antithetic partner $\omega_1$ is of length $m$: that is, $\mathbb{E}\left(\text{len}(\omega_2)|\text{len}(\omega_1) = m\right)$.

As a warm-up, consider the simpler marginal expected lengths. Note that

$$p\left(\text{len}(\omega) = m\right) = (1-p)^m p. \tag{46}$$

It follows that

$$\mathbb{E}\left(\text{len}(\omega)\right) = \sum_{m=0}^{\infty} m(1-p)^m p = \frac{1-p}{p} \tag{47}$$

where we evaluated the arithmetic-geometric series. We reported this result in Eq. 14. Meanwhile, the probability of a walk being of length $i$ given that its antithetic partner is of length $m$ is

$$p(\text{len}(\omega_2) = i|\text{len}(\omega_1) = m) = \begin{cases} \left(\frac{1-2p}{1-p}\right)^i \frac{p}{1-p} & \text{if } i < m, \\ 0 & \text{if } i = m, \\ \left(\frac{1-2p}{1-p}\right)^m (1-p)^{i-m-1} p & \text{if } i > m. \end{cases} \tag{48}$$

The analagous sum then becomes

$$\mathbb{E}\left(\text{len}(\omega_2)|\text{len}(\omega_1) = m\right) = \sum_{i=0}^{m-1} \left(\frac{1-2p}{1-p}\right)^i \frac{p}{1-p} i + \sum_{i=m+1}^{\infty} \left(\frac{1-2p}{1-p}\right)^m (1-p)^{i-m-1} pi. \tag{49}$$

After straightforward but tedious algebra, this evaluates to

$$\mathbb{E}\left(\text{len}(\omega_2)|\text{len}(\omega_1) = m\right) = \frac{1-2p}{p} + 2\left(\frac{1-2p}{1-p}\right)^m, \tag{50}$$

as stated in Eq. 15. Note that this is greater than $\mathbb{E}(\text{len}(\omega))$ when $m$ is small and smaller than $\mathbb{E}(\text{len}(\omega))$ when $m$ is large; the two walk lengths are negatively correlated.

### 8.4   On the superiority of q-GRFs (proof of Theorem 3.2)

Here, we provide a proof of the central result of Theorem 3.2: that the introduction of antithetic termination reduces the variance of estimators of the matrix $(\mathbf{I}_N - \mathbf{U})^{-2}$. From App. 8.2, all our results will trivially extend to the 2-regularised Laplacian kernel $\mathbf{K}_{\text{lap}}^{(2)}$.

We will begin by assuming that the graph is $d$-regular, that all edges have equal weights denoted $w$, and that our sampling strategy involves the random walker choosing one of its neighbours with equal probability at each timestep. We will relax these assumptions in App. 8.5.

We have seen that antithetic termination does not modify the walkers' marginal termination behaviour, so the variance of the estimator $\phi(i)^\top\phi(j)$ is only affected via the second-order term $\mathbb{E}\left[(\phi(i)^\top\phi(j))^2\right]$. Writing out the sums,

$$
\begin{aligned}
(\phi(i)^\top\phi(j))^2 = \frac{1}{m^4} \sum_{x,y\in\mathcal{N}} \sum_{k_1,l_1,k_2,l_2=1}^{m} \sum_{\omega_1\in\Omega_{ix}} \sum_{\omega_2\in\Omega_{jx}} \sum_{\omega_3\in\Omega_{iy}} \sum_{\omega_4\in\Omega_{jy}} \frac{\widetilde{\omega}(\omega_1)}{p(\omega_1)} \frac{\widetilde{\omega}(\omega_2)}{p(\omega_2)} \frac{\widetilde{\omega}(\omega_3)}{p(\omega_3)} \frac{\widetilde{\omega}(\omega_4)}{p(\omega_4)} \\
\cdot \mathbb{I}(\omega_1\in\bar{\Omega}(k_1,i))\mathbb{I}(\omega_2\in\bar{\Omega}(l_1,j))\mathbb{I}(\omega_3\in\bar{\Omega}(k_2,i))\mathbb{I}(\omega_4\in\bar{\Omega}(l_2,j)).
\end{aligned}
$$
(51)

To remind the reader: the variables $x,y$ sum over the nodes of the graph $\mathcal{N}$. $k_1$ and $l_1$ enumerate all the $m$ walks sampled out of node $i$, whilst $k_2$ and $l_2$ enumerate walks from $j$. The sum over $\omega_1\in\Omega_{ix}$ is over all possible walks between nodes $i$ and $x$. $\widetilde{\omega}(\omega_1)$ evaluates the product of edge weights traversed by the walk $\omega_1$, which is $w^{\mathrm{len}(\omega_1)}$ in the equal-weights case (with $\mathrm{len}(\omega_1)$ denoting the number of edges in $\omega_1$). $p(\omega_1)$ is the *marginal* probability of the prefix subwalk $\omega_1$, which is equal to $((1-p)/d)^{\mathrm{len}(\omega_1)}$ on a $d$-regular graph. Lastly, the indicator function $\mathbb{I}(\omega_1\in\bar{\Omega}(k_1,i))$ evaluates to 1 if the $k_1$th walk out of node $i$ (denoted $\bar{\Omega}(k_1,i)$) contains the walk $\omega_1$ as a prefix subwalk and 0 otherwise.

We immediately note that our scheme only ever correlates walks leaving the same node, so walks out of different nodes remain independent. Therefore,

$$
\begin{aligned}
\mathbb{E}\left[\mathbb{I}(\omega_1\in\bar{\Omega}(k_1,i))\mathbb{I}(\omega_2\in\bar{\Omega}(l_1,j))\mathbb{I}(\omega_3\in\bar{\Omega}(k_2,i))\mathbb{I}(\omega_4\in\bar{\Omega}(l_2,j))\right] \\
= p(\omega_1\in\bar{\Omega}(k_1,i),\omega_3\in\bar{\Omega}(k_2,i))p(\omega_2\in\bar{\Omega}(l_1,j),\omega_4\in\bar{\Omega}(l_2,j)).
\end{aligned}
$$
(52)

Consider the term in the sum corresponding to one particular set of walks $(k_1,l_1,k_2,l_2)$,

$$
\begin{aligned}
\sum_{x,y\in\mathcal{N}} \sum_{\omega_1\in\Omega_{ix}} \sum_{\omega_2\in\Omega_{jx}} \sum_{\omega_3\in\Omega_{iy}} \sum_{\omega_4\in\Omega_{jy}} \frac{\widetilde{\omega}(\omega_1)}{p(\omega_1)} \frac{\widetilde{\omega}(\omega_2)}{p(\omega_2)} \frac{\widetilde{\omega}(\omega_3)}{p(\omega_3)} \frac{\widetilde{\omega}(\omega_4)}{p(\omega_4)} \\
\cdot p(\omega_1\in\bar{\Omega}(k_1,i),\omega_3\in\bar{\Omega}(k_2,i))p(\omega_2\in\bar{\Omega}(l_1,j),\omega_4\in\bar{\Omega}(l_2,j)).
\end{aligned}
$$
(53)

This object will be of central importance and is referred to as the *correlation term*. In the sum over $k_1,k_2,l_1,l_2$, there are three possibilities to consider. We stress again that $k_{1,2}$ refers to a pair of walks out of node $i$ and $l_{1,2}$ refers to a pair out of $j$.

- **Case 1**, **same-same**, $k_1=k_2, l_1=l_2$: the pair of walks out of $i$ are identical and the pair of walks out of $j$ are identical. This term will not be modified by antithetic coupling since the marginal walk behaviour is unmodified and walks out of different nodes remain independent.

- **Case 2**, **different-different**, $k_1\neq k_2, l_1\neq l_2$: the walks out of both $i$ and $j$ differ, and each pair may be antithetic or independent. This term will be modified by the coupling.

- **Case 3**, **same-different**. $k_1=k_2, l_1\neq l_2$: the walks out of $i$ differ – and may exhibit antithetic or independent termination – but the walks out of $j$ are the same. This term will be modified by the coupling. Note that the $i$ and $j$ labels are arbitrary so we have chosen one ordering for concreteness.

If we can reason that the contributions from each of these possibilities $1-3$ either remains the same or is reduced by the introduction of antithetic coupling, then from Eq. 51 we can conclude that the entire sum and therefore the Laplacian kernel estimator variance is suppressed. For completeness, we

write out the entire sum from Eq. 51 with the degeneracy factors below:

$$(\phi(i)^\top \phi(j))^2 = \frac{1}{m^4} \sum_{x,y \in \mathcal{N}} \Big\{$$

$$\left. \begin{array}{l} m^2 \sum_{\omega_1 \in \Omega_{ix}} \sum_{\omega_2 \in \Omega_{jx}} \sum_{\omega_3 \in \Omega_{iy}} \sum_{\omega_4 \in \Omega_{jy}} \frac{\widetilde{\omega}(\omega_1)}{p(\omega_1)} \frac{\widetilde{\omega}(\omega_2)}{p(\omega_2)} \frac{\widetilde{\omega}(\omega_3)}{p(\omega_3)} \frac{\widetilde{\omega}(\omega_4)}{p(\omega_4)} \\ \cdot \mathbb{I}(\omega_1 \in \bar{\Omega}(k_1, i)) \mathbb{I}(\omega_2 \in \bar{\Omega}(l_1, j)) \mathbb{I}(\omega_3 \in \bar{\Omega}(k_1, i)) \mathbb{I}(\omega_4 \in \bar{\Omega}(l_1, j)) \end{array} \right\} \text{same-same (1)}$$

$$\left. \begin{array}{l} +m^2(m-1)^2 \sum_{\omega_1 \in \Omega_{ix}} \sum_{\omega_2 \in \Omega_{jx}} \sum_{\omega_3 \in \Omega_{iy}} \sum_{\omega_4 \in \Omega_{jy}} \frac{\widetilde{\omega}(\omega_1)}{p(\omega_1)} \frac{\widetilde{\omega}(\omega_2)}{p(\omega_2)} \frac{\widetilde{\omega}(\omega_3)}{p(\omega_3)} \frac{\widetilde{\omega}(\omega_4)}{p(\omega_4)} \\ \cdot \mathbb{I}(\omega_1 \in \bar{\Omega}(k_1, i)) \mathbb{I}(\omega_2 \in \bar{\Omega}(l_1, j)) \mathbb{I}(\omega_3 \in \bar{\Omega}(k_2, i)) \mathbb{I}(\omega_4 \in \bar{\Omega}(l_2, j)) \end{array} \right\} \text{different-different (2)}$$

$$\left. \begin{array}{l} +m^2(m-1) \sum_{\omega_1 \in \Omega_{ix}} \sum_{\omega_2 \in \Omega_{jx}} \sum_{\omega_3 \in \Omega_{iy}} \sum_{\omega_4 \in \Omega_{jy}} \frac{\widetilde{\omega}(\omega_1)}{p(\omega_1)} \frac{\widetilde{\omega}(\omega_2)}{p(\omega_2)} \frac{\widetilde{\omega}(\omega_3)}{p(\omega_3)} \frac{\widetilde{\omega}(\omega_4)}{p(\omega_4)} \\ \cdot \mathbb{I}(\omega_1 \in \bar{\Omega}(k_1, i)) \mathbb{I}(\omega_2 \in \bar{\Omega}(l_1, j)) \mathbb{I}(\omega_3 \in \bar{\Omega}(k_2, i)) \mathbb{I}(\omega_4 \in \bar{\Omega}(l_1, j)) \\ +m^2(m-1) \sum_{\omega_1 \in \Omega_{ix}} \sum_{\omega_2 \in \Omega_{jx}} \sum_{\omega_3 \in \Omega_{iy}} \sum_{\omega_4 \in \Omega_{jy}} \frac{\widetilde{\omega}(\omega_1)}{p(\omega_1)} \frac{\widetilde{\omega}(\omega_2)}{p(\omega_2)} \frac{\widetilde{\omega}(\omega_3)}{p(\omega_3)} \frac{\widetilde{\omega}(\omega_4)}{p(\omega_4)} \\ \cdot \mathbb{I}(\omega_1 \in \bar{\Omega}(k_1, i)) \mathbb{I}(\omega_2 \in \bar{\Omega}(l_1, j)) \mathbb{I}(\omega_3 \in \bar{\Omega}(k_1, i)) \mathbb{I}(\omega_4 \in \bar{\Omega}(l_2, j)). \Big\} \end{array} \right\} \text{same-different (3)}$$

$$(54)$$

We now address each case $1-3$ in turn.

### 8.4.1 Case 1: $k_1 = k_2, l_1 = l_2$

As argued above, case 1 is trivial. By design, antithetic termination does not affect the marginal walk behaviour (a sufficient condition for the estimator to remain unbiased). This means that it cannot affect terms that consider a single walk out of node $i$ and a single walk out of $j$, and all terms of case 1 are unchanged by the introduction of antithetic termination.

### 8.4.2 Case 2: $k_1 \neq k_2, l_1 \neq l_2$

Now we consider terms where both the walks out of node $i$ and the walks out of node $j$ differ. To emphasise, we are considering 4 different random walks: 2 out of $i$ and 2 out of $j$.

Within this setting, we will need to consider the situations where either i) one or ii) both of the pairs exhibit antithetic termination rather than i.i.d.. Terms of both kind will appear when we use ensembles of antithetic pairs. We need to check that in both cases the result is smaller compared to when both pairs are i.i.d..

To evaluate these terms, we first need to understand how inducing antithetic termination modifies the joint distribution $p(\omega_1 \in \bar{\Omega}(k_1, i), \omega_3 \in \bar{\Omega}(k_2, i))$: namely, the probability that two randomly sampled walks $\bar{\Omega}(k_1, i)$ and $\bar{\Omega}(k_2, i)$ contain the respective subwalks $\omega_1$ and $\omega_3$, given that their termination is either i.i.d. or antithetic. In the i.i.d. case, it is straightforward to convince oneself that

$$p(\omega_1 \in \bar{\Omega}(1, i), \omega_3 \in \bar{\Omega}(3, i)) = \left(\frac{1-p}{d}\right)^m \left(\frac{1-p}{d}\right)^n, \tag{55}$$

where $m$ and $n$ denote the lengths of subwalks $\omega_1$ and $\omega_3$, respectively. With antithetic termination, from Eq. 13 it follows that the probability of sampling a walk $\bar{\Omega}_3$ of length $j$ conditioned on sampling an antithetic partner $\bar{\Omega}_1$ of length $i$ is

$$p(\text{len}(\bar{\Omega}_3) = j | \text{len}(\bar{\Omega}_1) = i) = \begin{cases} \left(\frac{1-2p}{1-p}\right)^j \frac{p}{1-p} & \text{if } j < i, \\ 0 & \text{if } j = i, \\ \left(\frac{1-2p}{1-p}\right)^i (1-p)^{j-i-1} p & \text{if } j > i. \end{cases} \tag{56}$$

Using these probabilities, it is then straightforward but algebraically tedious to derive the joint probabilities over subwalks

$$p(\omega_1 \in \bar{\Omega}(1, i), \omega_3 \in \bar{\Omega}(3, i)) = \begin{cases} \frac{1}{d^{m+n}} \left(\frac{1-2p}{1-p}\right)^n (1-p)^m & \text{if } n < m, \\ \frac{1}{d^{2m}} (1-2p)^m & \text{if } n = m, \\ \frac{1}{d^{m+n}} \left(\frac{1-2p}{1-p}\right)^m (1-p)^n & \text{if } n > m, \end{cases} \tag{57}$$

where $m$ is the length of $\omega_1$, $n$ is the length of $\omega_3$ and $i$ is now the index of a particular node.

To be explicit, we have integrated over the conditional probabilities of *walks* of particular lengths $(i, j)$ to obtain the joint probabilities of sampled walks containing *subwalks* of particular lengths $(m, n)$. Let us consider the case of $n < m$ as an example. Using Eq. 56,

$$p(\omega_1 \in \bar{\Omega}(1, i), \omega_3 \in \bar{\Omega}(3, i)) = \frac{1}{d^{m+n}} \sum_{i=m}^{\infty} \left[ \sum_{j=n}^{i-1} \left( \frac{1-2p}{1-p} \right)^j \frac{p}{1-p} (1-p)^i p+ \right.$$
$$\left. + \sum_{j=i+1}^{\infty} \left( \frac{1-2p}{1-p} \right)^i (1-p)^{j-i-1} p (1-p)^i p \right], \tag{58}$$

where the branching factors of $d$ appeared because at every timestep the subwalks have $d$ possible edges to choose from. After we have completed the particular subwalks of lengths $m$ and $n$ we no longer care about *where* the walks go, just their lengths, so we stop accumulating these multiplicative factors. Computing the summations in Eq. 58 (which are all straightforward geometric series), we quickly arrive at the top line of Eq. 57.

Returning to our main discussion, note that in the $d$-regular, equal-weights case,

$$\sum_{\omega_1 \in \Omega_{ix}} \sum_{\omega_3 \in \Omega_{iy}} \frac{\widetilde{\omega}(\omega_1)}{p(\omega_1)} \frac{\widetilde{\omega}(\omega_3)}{p(\omega_3)} p(\omega_1 \in \bar{\Omega}(k_1, i), \omega_3 \in \bar{\Omega}(k_2, i))$$
$$= \sum_{\omega_1 \in \Omega_{ix}} \sum_{\omega_3 \in \Omega_{iy}} \left( \frac{wd}{1-p} \right)^{m+n} p(\omega_1 \in \bar{\Omega}(k_1, i), \omega_3 \in \bar{\Omega}(k_2, i)). \tag{59}$$

The summand depends only on walk lengths $m, n$ but not direction, which invites us to decompose the sum $\sum_{\omega_1 \in \Omega_{ix}} (\cdot)$ over walks between nodes $i$ and $x$ to a sum over walk lengths, weighted by the number of walks at each length. Explicitly,

$$\sum_{\omega_1 \in \Omega_{ix}} (\cdot) = \sum_{n=0}^{\infty} (\mathbf{A}^n)_{ix} (\cdot), \tag{60}$$

with $\mathbf{A}$ the (unweighted) adjacency matrix. We have used the fact that $(\mathbf{A}^n)_{ij}$ counts the number of walks of length $n$ between nodes $i$ and $x$. $\mathbf{A}$ is symmetric so has a convenient decomposition into orthogonal eigenvectors and real eigenvalues:

$$(\mathbf{A}^n)_{ix} = \sum_{k=1}^{N} \lambda_k^n \langle i|k \rangle \langle k|x \rangle \tag{61}$$

where $|k\rangle$ enumerates the $N$ eigenvectors of $\mathbf{A}$ with corresponding eigenvalues $\lambda_k$, and $\langle i|$ and $\langle x|$ are unit vectors in the $i$ and $x$ coordinate axes, respectively. We remind the reader that we have adopted Dirac's bra-ket notation; $|y\rangle$ denotes the vector $\boldsymbol{y}$ and $\langle y|$ denotes $\boldsymbol{y}^{\top}$.

Inserting Eqs 61 and 60 into Eq. 59 and using the probability distributions in Eq. 55 and 57, our all-important variance-determining correlation term from Eq. 53 evaluates to

$$\sum_{x,y \in \mathcal{N}} \sum_{k_1,k_2,k_3,k_4=1}^{N} B_{k_1,k_3}^{(i)} B_{k_2,k_4}^{(j)} \langle i|k_1 \rangle \langle k_1|x \rangle \langle j|k_2 \rangle \langle k_2|x \rangle \langle i|k_3 \rangle \langle k_3|y \rangle \langle j|k_4 \rangle \langle k_4|y \rangle, \tag{62}$$

where the matrix elements $B_{k_1,k_3}^{(i)}$ and $B_{k_2,k_4}^{(j)}$, corresponding to the pairs of walkers out of $i$ and $j$ respectively, are equal to one of the two following expressions:

$$B_{k_1,k_3} = \begin{cases} C_{k_1,k_3} := \frac{1}{1-w\lambda_{k_1}} \frac{1}{1-w\lambda_{k_3}} & \text{if i.i.d.} \\ D_{k_1,k_3} := \frac{1}{1-w\lambda_{k_1}} \frac{1}{1-w\lambda_{k_3}} \frac{1-w^2\lambda_{k_1}\lambda_{k_3}}{1-cw^2\lambda_{k_1}\lambda_{k_3}} & \text{if antithetic.} \end{cases} \tag{63}$$

Here, $c$ is a constant defined by $c := \frac{1-2p}{(1-p)^2}$ with $p$ the termination probability. These forms are straightforward to compute with good algebraic bookkeeping; we omit details for economy of space.

Eq. 62 can be simplified. Observe that $\sum_{x \in \mathcal{N}} |x\rangle \langle x| = \mathbf{I}_N$ ('resolution of the identity'), and that since the eigenvectors of $\mathbf{A}$ are orthogonal $\langle k_1|k_2 \rangle = \delta_{k_1,k_2}$. Applying this, we can write

$$\sum_{k_1,k_3=1}^{N} B_{k_1,k_3}^{(i)} B_{k_1,k_3}^{(j)} \langle i|k_1 \rangle \langle j|k_1 \rangle \langle i|k_3 \rangle \langle j|k_3 \rangle. \tag{64}$$

Our task is then to determine whether 64 is reduced by conditioning that either one or both of the pairs of walkers are antithetic rather than independent. That is,

$$\sum_{k_1=1}^{N}\sum_{k_3=1}^{N}\left(C_{k_1,k_3}D_{k_1,k_3}-C_{k_1,k_3}C_{k_1,k_3}\right)\langle i|k_1\rangle\langle j|k_1\rangle\langle i|k_3\rangle\langle j|k_3\rangle \overset{?}{\leq} 0, \qquad (65)$$

$$\sum_{k_1=1}^{N}\sum_{k_3=1}^{N}\left(D_{k_1,k_3}D_{k_1,k_3}-C_{k_1,k_3}C_{k_1,k_3}\right)\langle i|k_1\rangle\langle j|k_1\rangle\langle i|k_3\rangle\langle j|k_3\rangle \overset{?}{\leq} 0. \qquad (66)$$

Define a vector $\boldsymbol{y}\in\mathbb{R}^N$ with entries $y_p := \langle i|k_p\rangle\langle j|k_p\rangle$, such that its $p$th element is the product of the $i$ and $j$th coordinates of the $p$th eigenvector $\boldsymbol{k}_p$. In this notation, Eqs 65 and 66 can be written

$$\sum_{p=1}^{N}\sum_{q=1}^{N}\left(C_{pq}D_{pq}-C_{pq}C_{pq}\right)y_p y_q \overset{?}{\leq} 0, \qquad (67)$$

$$\sum_{p=1}^{N}\sum_{q=1}^{N}\left(D_{pq}D_{pq}-C_{pq}C_{pq}\right)y_p y_q \overset{?}{\leq} 0. \qquad (68)$$

For Eqs 67 and 68 to be true for arbitrary graphs, it is sufficient that the matrices $\mathbf{E}$ and $\mathbf{F}$ with matrix elements $E_{pq} := C_{pq}D_{pq}-C_{pq}C_{pq}$ and $F_{pq} := D_{pq}D_{pq}-C_{pq}C_{pq}$ are *negative definite*. Our next task is to prove that this is the case.

First, consider $\mathbf{E}$, where just one of the two pairs of walkers is antithetic. Putting in the explicit forms of $C_{pq}$ and $D_{pq}$ from Eq. 63,

$$E_{pq} = -\left(\frac{1}{(1-\bar{\lambda}_p)(1-\bar{\lambda}_q)}\right)^2\frac{p^2}{(1-p)^2}\frac{\bar{\lambda}_p\bar{\lambda}_q}{1-\frac{1-2p}{(1-p)^2}\bar{\lambda}_p\bar{\lambda}_q} \qquad (69)$$

where for notational compactness we took $\bar{\lambda}_p := w\lambda_p$ (the eigenvalues of the *weighted* adjacency matrix $\mathbf{U}$). Taylor expanding,

$$E_{pq} = -\left(\frac{1}{(1-\bar{\lambda}_p)(1-\bar{\lambda}_q)}\right)^2\bar{\lambda}_p\bar{\lambda}_q\frac{p^2}{(1-p)^2}\sum_{m=0}^{\infty}\left(\frac{1-2p}{(1-p)^2}\bar{\lambda}_p\bar{\lambda}_q\right)^m. \qquad (70)$$

Inserting this into Eq. 67, we get

$$\sum_{p=1}^{N}\sum_{q=1}^{N}E_{pq}y_p y_q = -\frac{p^2}{(1-p)^2}\sum_{m=0}^{\infty}\left(\sum_{p=1}^{N}\frac{\bar{\lambda}_p}{(1-\bar{\lambda}_p)^2}\left(\frac{\sqrt{1-2p}}{1-p}\bar{\lambda}_p\right)^m y_p\right)^2 \leq 0, \qquad (71)$$

which implies that $\mathbf{E}$ is indeed negative definite. Note that we have not made any additional assumptions about the values of $p$ and $w$ beyond those already stipulated: namely, $0 < p \leq \frac{1}{2}$ and $\bar{\lambda}_{\max} < 1$.

Next, consider $\mathbf{F}$, where both pairs of walkers are antithetic. Again inserting Eqs 63, we find that

$$F_{pq} = \left(\frac{1}{(1-\bar{\lambda}_p)(1-\bar{\lambda}_q)}\right)^2\left[\left(\frac{1-\bar{\lambda}_p\bar{\lambda}_q}{1-c\bar{\lambda}_p\bar{\lambda}_q}\right)^2-1\right] \qquad (72)$$

where we remind the reader that $c = \frac{1-2p}{(1-p)^2}$. The Taylor expansion in $\bar{\lambda}_p\bar{\lambda}_q$ is

$$F_{pq} = \left(\frac{1}{(1-\bar{\lambda}_p)(1-\bar{\lambda}_q)}\right)^2\left[\sum_{i=1}^{\infty}(\bar{\lambda}_p\bar{\lambda}_q)^i[(i+1)c^i - 2ic^{i-1} + (i-1)c^{i-2}]\right] \qquad (73)$$

$\mathbf{F}$ is *not* generically negative definite, but it will be at sufficiently small $p$ or $w$. Taylor expanding in $w$,

$$F_{pq} = 2(c-1)w^2\lambda_p\lambda_q + \mathcal{O}(w^3). \qquad (74)$$

The $\mathcal{O}(w^2)$ contribution is manifestly negative definite because $c < 1$. We cannot guarantee this for the higher order terms, but we can take the limit $w \to 0$ and treat them as a perturbation. It is

clear that the spectral radius of the perturbation will go to $0$ faster than the spectral radius of the leading term (all its contributions are proportional to higher powers of $w$), so it follows from Weyl's perturbation inequality [Bai et al., 2000] that $F_{pq}$ will indeed by negative definite provided $w$ is small enough. Therefore, at sufficiently small $w$ correlation terms with both pairs antithetic are suppressed as required.

Taylor expanding in $c \to 1$ (which corresponds to $p \to 0$) instead of $\lambda_p \lambda_q$, we can make exactly analogous arguments to find that $\mathbf{F}$ is also guaranteed to be negative definite with when $p$ is sufficiently small. Briefly: let $c = 1 - \delta$ with $\delta = \left( \frac{p}{1-p} \right)^2$. Then we have that

$$
\begin{aligned}
F_{pq} &= \left( \frac{1}{(1 - \bar{\lambda}_p)(1 - \bar{\lambda}_q)} \right)^2 \left( \left( \frac{1 - \bar{\lambda}_p \bar{\lambda}_q}{1 - \bar{\lambda}_p \bar{\lambda}_q + \delta \bar{\lambda}_p \bar{\lambda}_q} \right)^2 - 1 \right) \\
&= \left( \frac{1}{(1 - \bar{\lambda}_p)(1 - \bar{\lambda}_q)} \right)^2 \left( \frac{-2\delta \bar{\lambda}_p \bar{\lambda}_q}{1 - \bar{\lambda}_p \bar{\lambda}_q} + \mathcal{O}(\delta^2) \right).
\end{aligned}
\tag{75}
$$

Taylor expanding $\frac{1}{1 - \bar{\lambda}_p \bar{\lambda}_q}$, it is easy to see that the operator defined by the $\mathcal{O}(\delta)$ term of Eq. 75 is negative definite. This part will dominate over higher order terms (which are *not* in general negative definite) when $\delta$ is sufficiently small, guaranteeing the effectiveness of our mechanism on these terms.

This concludes our study of variance contributions in Eq. 53 where $k_1 \neq k_2$, $l_1 \neq l_2$. We have found that these correlation terms are indeed suppressed by antithetic termination when $p$ or $\rho(\mathbf{U})$ is small enough.

### 8.4.3   Case 3: $k_1 = k_2$, $l_1 \neq l_2$

We now consider terms where $k_1 = k_2$ and $l_1 \neq l_2$. We are considering a total of 3 walks: just 1 out of node $i$ but a pair (which may be antithetic or i.i.d.) out of node $j$. We inspect the term

$$
\sum_{\omega_1 \in \Omega_{ix}} \sum_{\omega_3 \in \Omega_{iy}} \left( \frac{wd}{1-p} \right)^{m+n} p(\omega_1 \in \bar{\Omega}(k_1, i), \omega_3 \in \bar{\Omega}(k_1, i)),
\tag{76}
$$

where $m$ denotes the length of $\omega_1$ and $n$ denotes the length of $\omega_3$. What is the form of $p(\omega_1 \in \bar{\Omega}(k_1, i), \omega_3 \in \bar{\Omega}(k_1, i))$? It is the probability that a *single* walk out of node $i$, $\bar{\Omega}(k_1, i)$, contains walks $\omega_1$ between nodes $i$ and $x$ and $\omega_3$ between $i$ and $y$ as subwalks. Such a walk must pass through all three nodes $i$, $x$ and $y$. After some thought,

$$
p(\omega_1, \omega_3 \in \bar{\Omega}(k_1, i)) = \begin{cases} \left( \frac{1-p}{d} \right)^m & \text{if } \omega_1 = \omega_3, \\ \left( \frac{1-p}{d} \right)^m & \text{if } \omega_3 \in \omega_1, \\ \left( \frac{1-p}{d} \right)^n & \text{if } \omega_1 \in \omega_3, \\ 0 & \text{otherwise.} \end{cases}
\tag{77}
$$

Here, $\omega_1 \in \omega_3$ means $\omega_1$ is a strict subwalk of $\omega_3$, so the sequence of nodes traversed is $i \to x \to y$. Likewise, $\omega_3 \in \omega_1$ implies a walk $i \to y \to x$. Summing these contributions,

$$
\begin{aligned}
\sum_{\omega_1 \in \Omega_{ix}} & \sum_{\omega_3 \in \Omega_{iy}} \left( \frac{wd}{1-p} \right)^{m+n} p(\omega_1 \in \bar{\Omega}(l_1, i), \omega_3 \in \bar{\Omega}(l_1, i)) \\
&= \underbrace{\sum_{\omega_1 \in \Omega_{ix}} \left( \frac{wd}{1-p} \right)^{2\text{len}(\omega_1)} p(\omega_1 \in \bar{\Omega}(k_1, i)) \delta_{xy}}_{\omega_1 = \omega_3, i \to x = y} \\
&+ \underbrace{\sum_{\omega_1 \in \Omega_{ix}} \left( \frac{wd}{1-p} \right)^{2\text{len}(\omega_1)} p(\omega_1 \in \bar{\Omega}(k_1, i)) \sum_{\omega_\delta \in \Omega_{xy}} \left( \frac{wd}{1-p} \right)^{\text{len}(\omega_\delta)} p(\omega_\delta \in \bar{\Omega}(k_1, x))}_{\omega_1 \in \omega_3, i \to x \to y} \\
&+ \underbrace{\sum_{\omega_3 \in \Omega_{iy}} \left( \frac{wd}{1-p} \right)^{2\text{len}(\omega_3)} p(\omega_3 \in \bar{\Omega}(k_1, i)) \sum_{\omega_\delta \in \Omega_{yx}} \left( \frac{wd}{1-p} \right)^{\text{len}(\omega_\delta)} p(\omega_\delta \in \bar{\Omega}(k_1, y))}_{\omega_3 \in \omega_1, i \to y \to x}.
\end{aligned}
\tag{78}
$$

We introduced $\omega_\delta$ for the sum over walks between nodes $x$ and $y$, and $p(\omega_\delta \in \bar{\Omega}(k_1, x))$ is the probability of some particular subwalk $x \to y$, equal to $\left(\frac{1-p}{d}\right)^{\text{len}(\omega_\delta)}$ in the $d$-regular case. $\omega_3$ is a dummy variable so can be relabelled $\omega_1$. The variance-determining correlation term from Eq. 53 becomes

$$
\begin{aligned}
\sum_{x,y \in \mathcal{N}} & \left[ \sum_{\omega_1 \in \Omega_{ix}} \left(\frac{wd}{1-p}\right)^{2\text{len}(\omega_1)} p(\omega_1 \in \bar{\Omega}(k_1, i)) \delta_{xy} \right. \\
& + \sum_{\omega_1 \in \Omega_{ix}} \left(\frac{wd}{1-p}\right)^{2\text{len}(\omega_1)} p(\omega_1 \in \bar{\Omega}(k_1, i)) \sum_{\omega_\delta \in \Omega_{xy}} \left(\frac{wd}{1-p}\right)^{\text{len}(\omega_\delta)} p(\omega_\delta \in \bar{\Omega}(k_1, x)) \\
& \left. + \sum_{\omega_1 \in \Omega_{iy}} \left(\frac{wd}{1-p}\right)^{2\text{len}(\omega_1)} p(\omega_1 \in \bar{\Omega}(k_1, i)) \sum_{\omega_\delta \in \Omega_{yx}} \left(\frac{wd}{1-p}\right)^{\text{len}(\omega_\delta)} p(\omega_\delta \in \bar{\Omega}(k_1, y)) \right] \\
& \qquad\qquad\qquad\qquad \cdot \sum_{k_2=1}^{N} \sum_{k_4=1}^{N} B_{k_2,k_4}^{(j)} \langle j|k_2 \rangle \langle k_2|x \rangle \langle j|k_4 \rangle \langle k_4|y \rangle .
\end{aligned}
\tag{79}
$$

where $B_{k_2,k_4}^{(j)}$ depends on whether the coupling of the pair of walkers out of node $j$ is i.i.d. or antithetic, as defined in Eq. 63. $x$ and $y$ are dummy variables so can also be swapped, and the sum over the walks $\omega_\delta$ is computed via the usual sum over walk lengths and eigendecomposition of $\mathbf{A}$. Using the resolution of the identity and working through the algebra, we obtain the correlation term

$$
\begin{aligned}
\sum_{x \in \mathcal{N}} & \left[ \sum_{\omega_1 \in \Omega_{ix}} \left(\frac{wd}{1-p}\right)^{2\text{len}(\omega_1)} p(\omega_1 \in \bar{\Omega}(k_1, i)) \right] \\
& \cdot \sum_{k_2,k_4=1}^{N} \left( \frac{1 - w^2 \lambda_{k_2} \lambda_{k_4}}{(1 - w\lambda_{k_2})(1 - w\lambda_{k_4})} \right) B_{k_2,k_4}^{(j)} \langle x|k_2 \rangle \langle k_2|j \rangle \langle x|k_4 \rangle \langle k_4|j \rangle .
\end{aligned}
\tag{80}
$$

Now observe that the prefactor in square brackets is positive for any node $x$ since it is the expectation of a squared quantity. This means that, for the sum in Eq. 80 to be suppressed by antithetic coupling, it is sufficient for the summation in its lower line to be reduced. Defining a vector $\boldsymbol{y} \in \mathbb{R}^N$ with elements $y_p := \langle x|k_p \rangle \langle k_p|j \rangle$, it becomes clear that we require that the operator $\mathbf{J}$ with matrix elements

$$
J_{pq} := \left( \frac{1 - w^2 \lambda_p \lambda_q}{(1 - w\lambda_p)(1 - w\lambda_q)} \right) (D_{pq} - C_{pq})
\tag{81}
$$

is negative definite. Using the forms in Eq. 63,

$$
J_{pq} = -\frac{w^2 \lambda_p \lambda_q}{(1 - w\lambda_p)^2 (1 - w\lambda_q)^2} \frac{\frac{p^2}{(1-p)^2}(1 - w^2 \lambda_p \lambda_q)}{1 - \frac{1-2p}{(1-p)^2} w^2 \lambda_p \lambda_q} .
\tag{82}
$$

Making very similar arguments to in Sec. 8.4.2 (namely, Taylor expanding and appealing to Weyl's perturbation inequality), we can show that, whilst this operator is not generically negative definite, it will be at sufficiently small $p$ or $w$.

This concludes the section of the proof addressing terms $k_1 = k_2$ and $l_1 \neq l_2$ (case 3). Again, these variance contributions are always suppressed by antithetic termination at sufficiently small $p$ or $\rho(\mathbf{U})$.

Having now considered all the possible variance contributions enumerated by cases $1 - 3$ and shown that each is either reduced or unmodified by the imposition of antithetic termination, we can finally conclude that our novel mechanism does indeed suppress the 2-regularised Laplacian kernel estimator variance for a $d$-regular graph of equal weights at sufficiently small $p$ or $\rho(\mathbf{U})$. $\qquad\square$

As mentioned in the main body of the manuscript, these conditions tend not to be very restrictive in experiments. Intriguingly, small $\rho(\mathbf{U})$ with $p = \frac{1}{2}$ actually works very well.

Our next task is to generalise these results to broader classes of graphs.

## 8.5 Extending the results to arbitrary graphs and sampling strategies (Theorem 3.2 cont.)

Throughout Sec. 8.4, we considered the simplest setting of a $d$-regular graph where all edges have equal weight. We have also taken a basic sampling strategy, with the walker choosing one of its

current node's neighbours at random at every timestep. Here we relax these assumptions, showing that our results remain true in more general settings.

### 8.5.1 Relaxing $d$-regularity

First, we consider graphs whose vertex degrees differ. It is straightforward to see that the terms in case 2 (Sec. 8.4.2) are unmodified because taking $d^m \to \prod_{i=1}^m d_i$ in $p(\omega_1)$ and $d^n \to \prod_{i=1}^n d_i$ in $p(\omega_3)$ is exactly compensated by the corresponding change in in joint probability $p(\omega_1 \in \bar{\Omega}(k_1, i), \omega_3 \in \bar{\Omega}(k_2, i))$. Our previous arguments all continue to hold.

Case 3 (Sec. 8.4.3) is only a little harder. Now the prefactor in square parentheses in the top line of Eq. 80 evaluates to

$$\left[ \sum_{\omega_1 \in \Omega_{ix}} \left( \frac{w}{1-p} \right)^{2\text{len}(\omega_1)} \left( \prod_{i=1}^{\text{len}(\omega_1)} d_i^2 \right) p(\omega_1 \in \bar{\Omega}(k_1, i)) \right] \tag{83}$$

which is still positive for any node $x$. The lower line of Eq. 80 is unmodified because once again the change $d^m \to \prod_{i=1}^m d_i$ exactly cancels in the marginal and joint probabilities, so $\mathbf{J}$ is unchanged and our previous conclusions prevail.

### 8.5.2 Weighted graphs

Now we permit edge weights to differ across the graph. Once again, case 2 (Sec. 8.4.2) is straightforward: instead of Eq. 60, we take

$$\sum_{\omega_1 \in \Omega_{ix}} \widetilde{\omega}(\omega_1) (\cdot) = \sum_{n=0}^{\infty} (\mathbf{U}^n)_{ix} (\cdot), \tag{84}$$

where $\mathbf{U}$ is the *weighted* adjacency matrix. We incorporate the product of each walk's edge weights into the combinatorial factor, then sum over walk lengths as before. In downstream calculations we drop all instances of $w$ and reinterpret $\lambda$ as the eigenvalues of the $\mathbf{U}$ instead of $\mathbf{A}$, but our arguments are otherwise unmodified; these variance contributions will be suppressed if $\rho(\mathbf{U})$ or $p$ is sufficiently small.

Case 3 (Sec. 8.4.3) is also easy enough; the bracketed prefactor of 80 becomes

$$\left[ \sum_{\omega_1 \in \Omega_{ix}} \left( \frac{1}{1-p} \right)^{2\text{len}(\omega_1)} \left( \prod_{i=1}^{\text{len}(\omega_1)} w_{i\sim i+1}^2 d_i^2 \right) p(\omega_1 \in \bar{\Omega}(k_1, i)) \right] \tag{85}$$

which is again positive. Here, $w_{i\sim i+1}$ denotes the weight associated with the edge between the $i$ and $i+1$th nodes of the walk. Therefore, it is sufficient that the matrix $\mathbf{J}$ with matrix elements

$$J_{pq} = - \frac{\lambda_p \lambda_q}{(1-\lambda_p)^2 (1-\lambda_q)^2} \frac{\frac{p^2}{(1-p)^2} (1 - \lambda_p \lambda_q)}{1 - \frac{1-2p}{(1-p)^2} \lambda_p \lambda_q} \tag{86}$$

is negative definite, with $\lambda_p$ now the $p$th eigenvalue of the *weighted* adjacency matrix $\mathbf{U}$. Following the same arguments as in Sec. 8.4.3, this will be the case at small enough $p$ or $\rho(\mathbf{U})$.

### 8.5.3 Different sampling strategies

Finally, we consider modifying the sampling strategy for random walks on the graph. We have previously assumed that the walker takes successive edges at random (i.e. with probability $\frac{1}{d_i}$), but the transition probability can also be a function of the edge weights. For example, if all the edge weights are positive, we might take

$$p(i \to j | \bar{s}) = \frac{w_{ij}}{\sum_{k \sim i} w_{ik}} \tag{87}$$

for the probability of transitioning from node $i$ to $j$ at a given timestep (with $w_{ij} \coloneqq \mathbf{U}_{ij}$), given that the walker does not terminate. This strategy increases the probability of taking edges with bigger

weights and which therefore contribute more to $(\mathbf{I}_N - \mathbf{U})^{-2}$ – something that empirically suppresses the variance on the estimator of the 2-regularised Laplacian kernel. Does antithetic termination reduce it further?

Case 2 (Sec. 8.4.2) is again easy; the $w$-dependent modifications to $p(\omega_1)$ and $p(\omega_3)$ are exactly compensated by adjustments to $p(\omega_1 \in \bar{\Omega}(k_1, i), \omega_3 \in \bar{\Omega}(k_2, i))$. To wit, Eq. 57 becomes

$$p(\omega_3 \in \bar{\Omega}(3, i), \omega_1 \in \bar{\Omega}(1, i)) = \begin{cases} \frac{\widetilde{\omega}(\omega_1)\widetilde{\omega}(\omega_3)}{\gamma(\omega_1)\gamma(\omega_3)} \left( \frac{1-2p}{1-p} \right)^n (1-p)^m & \text{if } n < m \\ \frac{\widetilde{\omega}(\omega_1)^2}{\gamma(\omega_1)^2} (1-2p)^m & \text{if } n = m \\ \frac{\widetilde{\omega}(\omega_1)\widetilde{\omega}(\omega_3)}{\gamma(\omega_1)\gamma(\omega_3)} \left( \frac{1-2p}{1-p} \right)^m (1-p)^n & \text{if } n > m. \end{cases} \quad (88)$$

where we defined a new function of a walk,

$$\gamma(\omega) := \prod_{i \in \omega} \sum_{k \in \mathcal{N}(i)} w_{ik}. \quad (89)$$

$\gamma$ computes the sum of edge weights connected to each node in the walk $\omega$ (excluding the last), then takes the product of these quantities. It is straightforward to check that, when all the graph weights are equal, $\frac{\widetilde{\omega}(\omega)}{\gamma(\omega)} = \frac{1}{d^m}$ with $m$ the length of $\omega$. Meanwhile, $p(\omega_1)$ becomes

$$p(\omega_1) = \frac{(1-p)^m \widetilde{\omega}(\omega_1)}{\gamma(\omega_1)} \quad (90)$$

such that these modifications cancel out when we evaluate Eq. 53.

Case 3 (Sec. 66) is also straightforward. The prefactor in square brackets is equal to 85 and is again positive for any valid sampling strategy $p(\omega_1 \in \bar{\Omega}(k_1, i))$ and $\mathbf{J}$ does not change, so our arguments still hold and these variance contributions are reduced by antithetic coupling.

We note that these arguments will generalise straightforwardly to any weight-dependent sampling strategy and are not particular to the linear case. $\widetilde{\omega}/\gamma$ can be replaced by some more complicated variant that defines a valid probability distribution $p(\omega_1 \in \bar{\Omega}(k_1, i))$ and antithetic termination will still prove effective.

### 8.5.4   Summary

In Sec. 8.5, our theoretical results for antithetic termination have proved robust to generalisations such as relaxing $d$-regularity and changing the walk sampling strategy. A qualitative explanation for this is as follows: upon making the changes, the ratio of the joint to marginal probablities

$$\frac{p(\omega_1, \omega_3)}{p(\omega_1)p(\omega_3)} \quad (91)$$

is unmodified. This is because *we know* how we are modifying the probability over walks and construct the estimator to compensate for it. Meanwhile, the correlations between walk *lengths* are insensitive to the walk directions, so in every case they continue to suppress the kernel estimator variance. The only kink is the terms described in Sec. 8.4.3 which require a little more work, but the mathematics conspires that our arguments are again essentially unmodified, though perhaps without such an intuitive explanation.

### 8.6   Beyond antithetic coupling (proof of Theorem 3.4)

Our final theoretical contribution is to consider random walk behaviour when TRVs are offset by *less* than $p$, $\Delta < p$. Unlike antithetic coupling, it permits simultaneous termination. Eqs 13 become

$$p(s_1) = p(s_2) = p, \quad p(\bar{s}_1) = p(\bar{s}_2) = 1 - p, \quad p(s_2|s_1) = \frac{p - \Delta}{p},$$

$$p(\bar{s}_2|s_1) = \frac{\Delta}{p}, \quad p(s_2|\bar{s}_1) = \frac{\Delta}{1-p}, \quad p(\bar{s}_2|\bar{s}_1) = \frac{1 - p - \Delta}{1 - p}. \quad (92)$$

The probability of two antithetic walks $\bar{\Omega}(1, i)$ and $\bar{\Omega}(3, i)$ containing subwalks $\omega_1$ and $\omega_3$ becomes

$$p(\omega_3 \in \bar{\Omega}(3,i), \omega_1 \in \bar{\Omega}(1,i)) = \begin{cases} \frac{1}{d^{m+n}} \left( \frac{1-p-\Delta}{1-p} \right)^n (1-p)^m & \text{if } n < m \\ \frac{1}{d^{2m}} (1-p-\Delta)^m & \text{if } n = m \\ \frac{1}{d^{m+n}} \left( \frac{1-p-\Delta}{1-p} \right)^m (1-p)^n & \text{if } n > m, \end{cases} \tag{93}$$

which the reader might compare to Eq. 57. In analogy to Eq. 63, this induces the matrix

$$D_{k_1,k_3}^\Delta := \frac{1}{1 - w^2 \lambda_{k_1} \lambda_{k_3} \frac{1-p-\Delta}{(1-p)^2}} \left( \frac{1 - w^2 \lambda_{k_1} \lambda_{k_3}}{(1 - w\lambda_{k_1})(1 - w\lambda_{k_3})} \right). \tag{94}$$

We can immediately observe that this is exactly equal to $C_{k_1,k_3}$ when $\Delta = p(1-p)$, so for a pair of walkers with this TRV offset the variance will be identical to the i.i.d. result. Replacing $D$ by $D^\Delta$ in $E_{pq}$ and $F_{pq}$ and $J_{pq}$ and reasoning about negative definiteness via their respective Taylor expansions (as well as the new possible cross-term $D_{k_1,k_3} D_{k_1,k_3}^\Delta$), it is straightforward conclude that variance is suppressed compared to the i.i.d. case provided $\Delta > p(1-p)$ and $\rho(\mathbf{U})$ or $p$ is sufficiently small. The $p \to 0$ limit demands a slightly more careful treatment: in order to stay in the regime $p(1-p) < \Delta < p$ we need to simultaneously take $\Delta \to 0$, e.g. by defining $\Delta(p) := p(1-p) + ap^2$ with the constant $0 < a < 1$. □

This result was reported in Theorem 3.4 of the main text.

## 8.7 What about diagonal terms?

The alert reader might remark that all derivations in Sec. 8.4 have taken $i \neq j$, considering estimators of the off-diagonal elements of the matrix $(\mathbf{I}_N - \mathbf{U})^{-2}$. In fact, estimators of the diagonal elements $\phi(i)^\top \phi(i)$ will be biased for both GRFs and q-GRFs if $\phi(i)$ is constructed using the same ensemble of walkers because each walker is manifestly correlated with, rather than independent of, itself. This is rectified by taking *two* ensembles of walkers out of each node, each of which may exhibit antithetic correlations among itself, then taking the estimator $\phi_1(i)^\top \phi_2(i)$. It is straightforward to convince oneself that, in this setup, the estimator is unbiased and q-GRFs will outperform GRFs. In practice, this technicality has essentially no effect on (q-)GRF performance and doubles runtime so we omit further discussion.

## 8.8 Further experimental details: compute, datasets and uncertainties

The experiments in Secs. 4.1, 4.2 and 4.4 were carried out on an Intel® Core™ i5-7640X CPU @ 4.00GHz × 4. Each required $\sim 1$ CPU hour. The experiments in Sec. 4.3 were carried out on a 2-core Xeon 2.2GHz with 13GB RAM and 33GB HDD. The computations for the largest considered graphs took $\sim 1$ CPU hour.

The real-world graphs and meshes were accessed from Ivashkin [2023] and Dawson-Haggerty [2023], with further information about the datasets available therein. Where we were able to locate them, the original papers presenting the graphs are: Zachary [1977], Lusseau et al. [2003], Newman [2006], Bollacker et al. [1998], Leskovec et al. [2007].

All our experiments report standard deviations on the means, apart from the clustering task in Sec. 4.3 because running kernelised $k$-means on large graphs is expensive.

