$$= w^4 (\lambda_p \lambda_q)^2 \sum_{i,j,k=0}^{\infty} (\lambda_p \lambda_q)^{i+j+k} w^{2i+j+k} (c-1) c^i (1 + c + i(c-1))(j+1)(k+1). \tag{56}$$

In fact, $\mathbf{F}$ is *not* generically negative definite, but will be at sufficiently small $p$ or $w$. Write $\mathbf{F} = w^4(\mathbf{G} + \mathbf{H})$, with

$$G_{pq} := (\lambda_p \lambda_q)^2 \left(c^2 - 1\right), \tag{57}$$

$$H_{pq} := (\lambda_p \lambda_q)^2 \sum_{i,j,k=0 \setminus \{i=j=k=0\}}^{\infty} (\lambda_p \lambda_q)^{i+j+k} w^{2i+j+k}(c-1)c^i(1+c+i(c-1))(j+1)(k+1). \tag{58}$$

$\mathbf{G}$ is manifestly negative definite because $c < 1$ but $\mathbf{H}$ may not be. Treat $\mathbf{H}$ as a perturbation to $\mathbf{G}$.

Recalling that the spectral radius of $\mathbf{H}$ is defined

$$\rho(\mathbf{H}) := \max_{\|\boldsymbol{x}\|_2 = 1} \mathbf{H}\boldsymbol{x}, \tag{59}$$

it is clear that the spectral radius of $\mathbf{H}$ approaches $0$ smoothly as $w \to 0$ since all its matrix elements vanish. Recall also an important corollary of Weyl's perturbation inequality: any perturbed eigenvalue of $\mathbf{F} + \mathbf{G}$ will be within one spectral radius $\rho(\mathbf{G})$ of the original eigenvalue of $\mathbf{F}$. This means that, by reducing $w$, we can shrink the spectral radius of $\mathbf{G}$ until $\rho(\mathbf{G}) < (\lambda_p \lambda_q)^2 \left(1 - c^2\right)$, at which point we are guaranteed that $\mathbf{F}$ will be negative definite. Hence, at sufficiently small $w$, correlation terms with both pairs antithetic are suppressed as required.

Taylor expanding in $c \to 1$ (which corresponds to $p \to 0$) instead of $\lambda_p \lambda_q$, we can make exactly analogous arguments to find that $\mathbf{F}$ is also guaranteed to be negative definite with when $p$ is sufficiently small. Briefly: let $c = 1 - \delta$ with $\delta = \left(\frac{p}{1-p}\right)^2$. Then we have that

$$\begin{aligned} F_{pq} &= \left(\frac{\bar{\lambda}_p \bar{\lambda}_q}{(1-\bar{\lambda}_p)(1-\bar{\lambda}_q)}\right)^2 \left((1-\delta)^2 \left(\frac{1 - \bar{\lambda}_p \bar{\lambda}_q}{1 - \bar{\lambda}_p \bar{\lambda}_q + \delta \bar{\lambda}_p \bar{\lambda}_q}\right)^2 - 1\right) \\ &= \left(\frac{\bar{\lambda}_p \bar{\lambda}_q}{(1-\bar{\lambda}_p)(1-\bar{\lambda}_q)}\right)^2 \left(\frac{-2\delta}{1 - \bar{\lambda}_p \bar{\lambda}_q} + \mathcal{O}(\delta^2)\right). \end{aligned} \tag{60}$$

Taylor expanding $\frac{1}{1-\bar{\lambda}_p \bar{\lambda}_q}$, it is easy to see that the operator defined by the $\mathcal{O}(\delta)$ term of Eq. 60 is negative definite. This part will dominate over higher order terms (which are *not* in general negative definite) when $\delta$ is sufficiently small, guaranteeing the effectiveness of our mechanism on these terms.

As an aside, we also note that Taylor expanding about $c = 0$ (which corresponds to $p \to \frac{1}{2}$) yields

$$F_{pq} = \left(\frac{\bar{\lambda}_p \bar{\lambda}_q}{(1-\bar{\lambda}_p)(1-\bar{\lambda}_q)}\right)^2 \left(-1 + \mathcal{O}(c^2)\right) \tag{61}$$

which is manifestly negative definite at small enough $c$. Hence, intriguingly, the $k_1 \neq k_2$ variance contributions are also suppressed in the $p \to \frac{1}{2}$ limit.