# OpenReview forum: "Quasi-Monte Carlo Graph Random Features"
_NeurIPS.cc/2023/Conference — NeurIPS 2023 spotlight_

### Official Review · Reviewer_qCjY · 2023-07-06

**Soundness:** 4 excellent
**Presentation:** 3 good
**Contribution:** 3 good
**Rating:** 7
**Confidence:** 3

**Summary:**

Kernel methods are important in many ML applications, but they suffer from scalability issues. In the Euclidean setting, random features methods (Rahimi & Recht NeurIPS 2017) can be used to "sketch" the kernel matrices and speed up computations. In the graph setting, however, no such random feature seems to have been available until a 2023 preprint by Choromanski.

The present paper present an improvement over Choromanski's method. To explain it, note that the basic idea in the 2023 preprint is to consider random features for the regularized graph Laplacian, and to obtain an approximation of an operator of the form $(I-U)^{-d}$ via importance-weighted random walks. These random walks have geometrically distributed lengths.

The main theoretical contribution of the present paper can now be explained: a simple modification of Choromanski's method, where the the geometrics of pairs of random walks become correlated, has the same zero bias and a smaller variance than the above method.
This requires an elementary, but long proof with a long case analysis.

Experimentally, the gains observed were:

* visible, but not too big for approximation of the kernel;
* impressive for graph diffusions (which is natural because this requires exponentiating the Laplacian);
* fairly significant for clustering and node attribute prediction methods.

**Strengths:**

The paper is clear and seemingly original. In general it is fairly well-written. Moreover, the "anthitetic" coupling is a simple little idea that could be applicable more broadly.



**Weaknesses:**

The present paper has been submitted before Choromanski's preprint appeared. As a result, applications of graph kernel random sketches are quite incipient still. It is hard to say whether the present paper will have a significant direct impact. _[Edit on 08/11: this was partly addressed in the rebuttal.]_

The proof of variance reduction is very long and difficult to check. (Though I believe it.)  (An "actionable version" of this point is that it would pay off to make this paper a bit less dependent on Chorumovski's preprint by eg. adding more experiments or theory.)


_Less important comments_

The notation $t_{1,2}$ is a bit weird as it represents a pair of RVs. Moreover, if I understand correctly, these r.v.s are resampled at each step of the walk.

In terms of exposition, it seems that all the authors need to do is to correlate the coin flips for the two walks at each step. Could you write their joint distribution purely in terms of the probabilities of 00, 01, 10 and 11?

**Questions:**

Can you point to more specific and realistic applications where this methodology would be useful? (A somewhat speculative discussion would be nice.)

--

Edit (08/11) -- addressed in rebuttal.

**Limitations:**

These are properly discussed.

---

> ### Author Rebuttal · Authors · 2023-08-08
>
> We thank the reviewer for reading the manuscript, and are delighted that they note the work’s novelty, clarity and strong experimental contributions. We agree that it is remarkable that such a conceptually simple technique to correlate walker lengths can lead to such rich behaviour and demand such a long proof. We hope the brief proof sketch will help explain the high-level strategy to the interested reader.
>
> We also agree that antithetic coupling is of broader interest for algorithms sampling random walks with geometrically distributed lengths. In fact, prompted by such comments from the reviewers, we have since implemented antithetic termination for numerical approximation of the PageRank vector, a popular measure of the importance of a node in a graph (https://www.jstor.org/stable/40232890). We find very large gains from our QMC scheme, reducing the $L_2$-norm error on the vector’s estimation by a factor of approximately 2. This further demonstrates the power and generality of our new sampling strategy. Plots are included in the attached PDF.
>
> We now address the reviewer’s questions and comments in detail.
>
> 1. *Relationship with Choromanski’s paper*: the reviewer correctly notes that the field of graph kernel random sketches is young and the impact of these mechanisms will become clearer with time. That said, we note that since submitting this draft Choromanski’s original preprint has been accepted to ICML with an oral presentation, suggesting that these scalable GRF techniques — and by extension their more efficient QMC variants — might be of interest to the community. The present manuscript also goes well beyond Choromanski’s work in many ways, including: considering a *quasi*-Monte Carlo scheme, using (q-)GRFs to approximate the heat kernel (Secs 2.1 and 4.1) and providing detailed theoretical analysis of both the old and new schemes (Appendices 8.3-8.6). Indeed, the $d$-regularised Laplacian and heat kernels are possibly the two most popular classes of kernels defined on graph nodes, listed as top examples in Eqs 17 and 18 Smola and Kondor’s seminal paper (https://people.cs.uchicago.edu/~risi/papers/SmolaKondor.pdf). We therefore anticipate that these new schemes to efficiently approximate them will prove very impactful. We will emphasise this in the manuscript.
> Moreover, we stress that antithetic termination is at its heart a scheme to sample more diverse random walks. As such, its applicability extends beyond the (nonetheless important) application of q-GRFs. This is made clear by our new PageRank results. We are eager to hear of any further suggestions for possible applications.
> 2. *Notation for termination random variables (TRVs)*: we thank the reviewer for their stylistic comment and will clarify the notation in the manuscript. They are correct that the TRVs are resampled at every timestep — an important point that we will highlight.
> 3. *Joint distribution over coin flips*: we thank the reviewer for their interesting comment. In Eq. 13 we opted to present the conditional probabilities. From here it is trivial to obtain the joint probabilities, but we agree that explicitly including them might make the exposition clearer. We have updated the manuscript.
> 4. *Applications where this methodology might be useful*: to quote Smola and Kondor’s seminal paper  (https://people.cs.uchicago.edu/~risi/papers/SmolaKondor.pdf), ‘since kernel-based algorithms, such as Support Vector Machines, Gaussian Processes, Kernel PCA, etc. capture the structure of $\mathcal{X}$ via the kernel $K : \mathcal{X} \times \mathcal{X} \to \mathbb{R}$, as long as we can define an appropriate kernel on our discrete input space, these algorithms can be imported wholesale, together with their error analysis, theoretical guarantees and empirical success’. Since any graph-based learning algorithm using the $d$-regularised and heat kernels can benefit from our efficient q-GRF construction, it follows that our techniques will enjoy applications in a very wide variety of downstream graph-based learning tasks. To give some concrete examples: inferring protein function (https://academic.oup.com/bioinformatics/article/21/suppl_1/i47/202991), community detection (https://arxiv.org/abs/1403.3148) and recommender systems (https://link.springer.com/chapter/10.1007/11731139_28). More recently, the heat kernel has been used in settings as diverse as manifold learning for deep generative modelling (https://arxiv.org/abs/2010.01761) and for solving single- and multiple-source shortest path problems (https://arxiv.org/abs/2010.01761). Speculatively, these kernels might also be used to modulate the regular attention mechanism in graph-based Transformers, incorporating topological signal and endowing a structural inductive bias (https://arxiv.org/abs/2106.05234, https://arxiv.org/abs/1710.10903, https://arxiv.org/abs/2107.07999). For big graphs where exact kernel evaluation is expensive, we expect that our efficient algorithm will prove especially impactful.
> Beyond kernel estimation, we have argued that antithetic termination is likely to be effective in any Monte Carlo algorithm sampling random walks with geometrically distributed lengths. Our new results confirm that this is the case for numerical estimation of the PageRank vector, immediately opening up diverse applications from ranking websites to analysing social networks.
> We agree that it is important to properly motivate the work and will add this discussion to the draft.
>
> We again sincerely thank the reviewer and invite them to respond with any further questions.

---

> > ### Comment · Reviewer_qCjY · 2023-08-11
> > **Thank you**
> >
> > Thank you for your response. I have raised my score slightly. I will look carefully at the discussion with other reviewers to see if any further changes to my score make sense.

---

> > > ### Author Response · Authors · 2023-08-14
> > > **Thank you and further update**
> > >
> > > We thank the reviewer for their prompt response and for raising their score.
> > >
> > > For the reviewer's interest and as further evidence of the applicability of antithetic termination beyond Choromanski's GRF setting, **we have now finished a proof that antithetic termination suppresses the variance on estimates of the PageRank vector for any graph**. This explains the excellent empirical results sent earlier in the rebuttal. We have added these new theoretical insights to the manuscript and are excited about further future applications in algorithms using terminating random walkers.

---

> > > > ### Comment · Reviewer_qCjY · 2023-08-17
> > > > **Thanks again**
> > > >
> > > > Upon reflection, it makes sense to once again raise my score by one point.

---

### Official Review · Reviewer_mBGP · 2023-07-09

**Soundness:** 3 good
**Presentation:** 3 good
**Contribution:** 3 good
**Rating:** 7
**Confidence:** 4

**Summary:**

The author(s) extend the work of Choromanski [2023] and introduce quasi monte carlo graph random features.
To estimate the 2-regularized Laplacian kernel, they now use random walk features for each node which have coupled probabilities of terminating after a certain number of steps.
This increases diversity in the features and interestingly leads to provable reduction of variance of the global estimates.
An empirical evaluation supports the theoretical findings by improved practical variance, as well as in a clustering and supervised node learning scenario.


**Strengths:**


The paper is well-written and mostly easy to follow.
While proofs are moved to the appendix, a sketch of proof of the main result is included in the paper. I like that.

The introduced way of increasing diversity in random walks is (up to my knowledge) novel and very easy to implement. While it is evaluated in the context of estimating the 2-regularized laplacian of a graph, I agree with the authors that this method is of independent interest.
An interesting question is if this kind of sampling improves results of other random walk based approaches, as well, or if the parctical impact is restricted to just a few scenarios (GRF being one of them).
As a result, I see potential impact for further research beyond this immediate application.



**Weaknesses:**

The introduction is structurally very similar to that of Choromanski [2023]. I'm assuming now that I know one of the authors of this submission. Otherwise, I would recommend the authors to revise the introduction to decrease similarity.


**Questions:**


The last sentence of the proof sketch seems to encompass quite some work. I suggest it to be extended to mention at least what kind of matrix is supposed to be negative definite.

Suggestions
l. 107: 'the the'
l. 164: 'the event that event that'
l. 207 and l. 222: 'path' should probably be 'walk'
References: I suggest the author(s) revise the references for consistent formatting. E.g. Arxiv vs Corr.Abs., as another example, Choromanski [2023] does not list any venue, at all.


**Limitations:**

The authors provide a dedicated section discussing potential (far away) societal impacts, as well as open questions with the current approach.
The discussion is adequate.

---

> ### Author Rebuttal · Authors · 2023-08-08
>
> We thank the reviewer for their careful reading of the manuscript and thoughtful feedback. We are delighted that they note this novel mechanism to increase the diversity of random walks is simple to implement and offers strong theoretical guarantees and empirical performance – in fact not only for the $2$-regularised Laplacian but also the $d$-regularised Laplacian (Sec. 8.1) and the heat kernel (Secs 2.1 and 4.1). They note that the method might be of independent interest for other random walk-based algorithms,  a possibility that we expect to dedicate substantial future work to. Prompted by comments from another reviewer, we have already implemented a new experiment applying antithetic termination to numerical estimation of the PageRank vector (a popular measure of a node’s importance in a graph – https://www.jstor.org/stable/40232890). We observe very large gains, showing the generality of the method beyond graph kernel estimation. Please see the attached PDF. It seems this simple scheme to improve random walk sampling efficiency is of very broad scope.
>
> We address the reviewer’s questions and suggestions in detail below.
>
> 1. *Structure of introduction*: we thank the reviewer for their observation, but will refrain from further comment in the interests of preserving anonymity.
> 2. *Final sentence of proof sketch*: the reviewer correctly observes that a key ingredient of our theoretical work is identifying a particular matrix that must be negative definite in order that antithetic termination suppresses the kernel estimator variance, and using Weyl’s perturbation inequality to identify regimes in which this is the case. As they say, this belies substantial technical work and, whilst the proof sketch is not intended to reproduce every detail, we agree that supplementing the discussion might benefit the motivated reader. We have updated the manuscript to include this.
> 3. *Typos and reference formatting*: we thank the reviewer for their careful observations and have corrected these minor mistakes in the manuscript.
>
> We again thank the reviewer for their close reading of the manuscript and helpful suggestions.

---

> > ### Comment · Reviewer_mBGP · 2023-08-18
> >
> > Thank you for your reply. My questions have been addressed. I like the authors extension to the computation of page rank scores. I believe this indeed shows that antithetic termination is a technique that can be relevant in many scenarios. I hence have increased my confidence and stand by my suggestion to accept this paper.

---

> > > ### Author Response · Authors · 2023-08-18
> > > **Thanks**
> > >
> > > We thank the reviewer for their response and for increasing their confidence score.

---

### Official Review · Reviewer_SAEb · 2023-07-27

**Soundness:** 3 good
**Presentation:** 3 good
**Contribution:** 3 good
**Rating:** 5
**Confidence:** 3

**Summary:**

This work proposes an efficient random-walk sampling approach that accelerates the estimation of the feature mapping for 2-regularized Laplacian kernels. The random walk sampling is based on antithetic termination, which is a variance reduction technique and sounds novel  when being applied to the estimation of graph random features. The technique sounds solid. The authors also conducted experiments from different angles to verify the effectiveness of the proposed approach in practice.

**Strengths:**

1. Using random walk with antithetic termination to estimate graph random features is novel, to the best knowledge of me.
2. Although I did not check the appendix, the technique and the benefits it may bring look reasonable to me.
3. The authors did a great job to explain the key insights behind. The paper is very well written.
4. The experiments are extensive. The benefits of the proposed approach is justified from various angles.

**Weaknesses:**

1. The biggest weakness of the work is the limited scope of this work. It is unclear why the 2-regularized Laplacian kernel is important and why we need to compute it. It seems that 1-regularized Lap kernel can be efficient computed by Pagerank algorithms, why we may need 2-regularized Lap kernel.
2. It is unclear how much the estimation of the 2-regularized Laplacian kernel relies on the random walk with antithetic termination. Can this random walk approach be only applied to 2-regularized Laplacian kernels or a broader range of graph kernels? This question is kind of related to whether the paper has over-claimed the contributions. Neither the title nor the introduction reflect that the improvement is only for  2-regularized Laplacian kernels, and they claim a broader range of contributions. I expect the authors to make the actual contributions more clear in the revised version.

**Questions:**

1. Why is the 2-regularized Laplacian kernel important? Adding experiments to compare with other graph kernels (in the aspects of both inference accuracy and computation) may improve the draft.
2. Is the random walk approach only applied to the estimation of 2-regularized Laplacian kernel?

---

> ### Author Rebuttal · Authors · 2023-08-08
>
> We sincerely thank the reviewer for their encouraging comments, especially pertaining to the novelty of the mechanism, the quality of the writing and the exhaustive nature of the experimental section. Below, we address minor points of misunderstanding and answer the reviewer’s questions.
>
> 1. *Scope of the work*: though it is our predominant focus, **the scope of antithetic termination is not limited to estimation of the $2$-regularised Laplacian**. As we show explicitly in Sec. 8.1, GRFs (and, by extension, our more efficient q-GRF variant) can be used to construct scalable estimates to $d$-regularised Laplacians for arbitrary integer $d$, covering a broad family of very popular graph kernels. Moreover, in Sec. 2.1 we show how these techniques can be used to estimate the heat kernel $K_\textrm{diff} = \textrm{exp}(- \widetilde{\textbf{L}} t)$ – a different graph kernel which is also widely-used – and in Table 1 we see that here the gains from employing antithetic termination are substantial. This covers what are often considered the two most popular classes of graph kernels, listed as the top examples in Eqs 17 and 18 of Smola and Kondor’s seminal paper (https://people.cs.uchicago.edu/~risi/papers/SmolaKondor.pdf). With strong physical motivation and practical efficacy, they are well-documented to enjoy widespread applications in any algorithms employing kernel-based algorithms on graphs, e.g. SVMs, Gaussian Processes, and kernelised PCA (e.g. https://academic.oup.com/bioinformatics/article/21/suppl_1/i47/202991, https://arxiv.org/abs/1403.3148, https://link.springer.com/chapter/10.1007/11731139_28). Exciting recent work also suggests that they might also be used to modulate regular attention to provide a structural inductive bias in graph-based Transformers (https://arxiv.org/abs/2106.05234, https://arxiv.org/abs/1710.10903, https://arxiv.org/abs/2107.07999). We hope the reviewer will agree that this demonstrates the broad scope of q-GRFs for kernel estimation, and will be sure to make this point clearer in the manuscript. What’s more, we would like to stress that **antithetic termination is fundamentally a procedure to sample more diverse discrete random walks which can be straightforwardly applied in Monte Carlo schemes beyond graph kernel estimation**. This further extends our work’s scope. We discuss this in detail later in the rebuttal (see point 4).
>
> 2. *Experimental comparison of graph kernels*: we again stress that the manuscript already includes experimental and theoretical discussion of different types of kernel beyond the $2$-regularised Laplacian (see Secs 2.1, 4.1 and 8.1). However, the purpose of the paper is not to compare how they perform on particular learning tasks, but rather present a general QMC scheme that can better the estimation of any one of them. Concretely, we are less interested in whether e.g. the $1$- or $2$-regularised Laplace kernel is better for node attribute prediction on trimesh graphs, but rather in whether approximating either of them with our new QMC scheme can improve efficiency and performance compared to the vanilla MC mechanism.
>
> 3. *Contributions, title and introduction*: as explained above, GRFs can be used for scalable estimation of a broad range  of kernels beyond the $2$-regularised Laplacian, and our theoretical and experimental results show that in every case our new q-GRFs variant converges better. Whilst we anticipate that the popular $d=2$ case might be of most direct interest, we agree that this is an important point and will emphasise it.
> Our paper’s title uses standard terminology: Choromanski coined ‘GRFs’ to describe their new technique for estimation of graph-based kernels, and ‘q-GRFs’ is a natural choice for our more efficient QMC variant. As early as the third sentence of the abstract we state that our chief interest will be the $2$-regularised Laplacian. We are transparent about this fact. That said, we again stress that the method’s applicability is more general: it is at its heart a technique to improve sampling on graphs.
> 4. *How are (q-)GRFs related to PageRank*? We sincerely thank the reviewer for referring us to PageRank, a popular measure of the importance of the nodes of a network. The PageRank vector is the steady state distribution of a particular random walk process, or equivalently the top eigenvector of a $1$-regularised Laplacian kernel. The reviewer correctly points out that efficient MC algorithms exist to numerically approximate this vector, but the scope of GRFs (and the q-GRFs variant we present) is broader: they provide a means to approximate the entire $d$-regularised Laplace kernel for arbitrary integer $d$, rather than just its largest eigenvector when $d=1$.
> That said, we agree that the literature on PageRank presents interesting connections to our work. Especially relevant is Alg. 1 of Avrachenkov’s ‘Monte Carlo Methods in PageRank Computation’ (https://www.jstor.org/stable/40232890), which samples $K$ random walks out of each node with probability of termination $p$ at every timestep and records where each walk ends. It is straightforward to incorporate antithetic termination into Avrachenkov’s algorithm, pairing up RWs and using our mechanism to anticorrelate their lengths. **Following the reviewer’s comment, we implemented this experiment and observed large gains: for a fixed number of walkers, antithetic termination  suppresses the $L_2$-norm error on the approximation of the PageRank vector by a factor of approximately 2** (see the attached PDF). We have added this remarkable result to the manuscript and will include accompanying theoretical discussion in the final draft. This further demonstrates the scope of antithetic termination beyond graph-based kernels.
>
> We again thank the reviewer for their helpful comments and questions. We hope the above clarifications and additions will be sufficient to allay any concerns and invite them to reach out with further questions.

---

> > ### Comment · Reviewer_SAEb · 2023-08-11
> > **Thanks for the response**
> >
> > I appreciate the authors' detailed response. I agree that antithetic termination is fundamentally a procedure to sample more diverse discrete random walks, which is a great contribution. The empirical observation on PageRank approximation is also interesting. I still have questions about how wide the range of graph kernels could be where the proposed approach may be applied.
> >
> > 1. "As we show explicitly in Sec. 8.1,..."
> >
> > Sorry, I did not see Sec. 8.1. Do you mean Sec. 2.1?
> >
> > 2. "construct scalable estimates to regularized Laplacians for arbitrary integer $d$, ..."
> >
> > I do not think the paper given its current content shows a general applicability of the proposed approach. The analysis is mainly on 2-regularized Laplacians if I did not miss anything. Also, I know that heat kernel has a wide range of applications, but I did not see how the proposed approach can estimate the heat kernel. The discussion on heat kernel in Sec 2.1 is just to motivate d-regularized Laplacians where 2-regularized Laplacian is actually what to approximate in this work. Please correct me if I am wrong.

---

> > > ### Author Response · Authors · 2023-08-12
> > > **Application to other graph kernels**
> > >
> > > We thank the reviewer for their reply.
> > >
> > > Estimates of the $2$-regularised Laplacian kernel $\mathbf{K}_L^{(2)}$ can be used to construct estimates to the $d$-regularised Laplacian $\mathbf{K}_L^{(d)}$  and heat kernels $\mathbf{K}_D$ (note the slight change in notation compared to the manuscript because of markdown compiler problems).
> > >
> > > For the $d$-regularised Laplacian kernel, the section of interest is 7.1 from the supplementary material: `on the approximation of the $d$-regularised Laplacian using GRFs' (with apologies to mistakenly referring to Sec. 8.1 after using a newer version of the draft). We summarise it briefly for the reviewer's convenience. As they correctly point out, we dedicate time in the manuscript to approximating efficiently $\mathbf{K}_L^{(2)} \coloneqq (\mathbf{I} - \mathbf{U})^{-2}$. By also multiplying by $\mathbf{I} - \mathbf{U}$, it is trivial to obtain  $\mathbf{K}_L^{(1)} \coloneqq (\mathbf{I} - \mathbf{U})^{-1}$. We can then take multiplicative combinations of (independent) estimators for an unbiased approximation to $\mathbf{K}_L^{(d)} \coloneqq (\mathbf{I} - \mathbf{U})^{-d}, d \in \mathbb{N}$. Since antithetic termination suppresses the variance on $\mathbf{K}_L^{(2)}$ and $\mathbf{I} - \mathbf{U}$ is known exactly, it follows that such estimates of ${\mathbf{K}_L^{(d)}}$ will enjoy lower variance. We will make this clearer in the manuscript.
> > >
> > > The argument for the heat kernel is very similar. Indeed, we note that `simulation of diffusion on a graph' (introduced in Sec. 2.1 and tested in Sec. 4.2) is exactly equivalent to approximating the action of the heat kernel on a vector corresponding to the initial state. To briefly summarise,
> > >
> > > $\mathbf{K}_D= \text{exp}(- \widetilde{\mathbf{L}}t) \simeq \left [(\mathbf{I}_N + \frac{t}{N_t} \widetilde{\mathbf{L}})^{-2}\right]^\frac{N_t}{2}=(\mathbf{K}_L^{(2)})^\frac{N_t}{2}$
> > >
> > > with equality becoming exact in the limit of a large number of discretisation time steps $N_t$. This is shown in Eq. 21 of the main text. In Sec. 4.2 we probe the quality of this approximation by comparing the error on $\mathbf{u}_t \coloneqq {\mathbf{K}}_D \mathbf{u}_0$ when ${\mathbf{K}}_D$ is estimated with GRFs and q-GRFs, and find it to be substantially improved in the latter case (see Table 1). It is also clear from the expression above that $\mathbf{K}_D$ is a special case of $\mathbf{K}_L^{(d)}$ in the large $d = N_t$ limit with $\mathbf{U}$ scaled appropriately, so theoretical guarantees follow.
> > >
> > > We hope this clarifies the concrete applicability of the approach to other kernels, and how q-GRFs are theoretically and empirically found to provide better estimates. We will try to make this more obvious in the manuscript.
> > >
> > > We are also pleased to report that the new PageRank application admits straightforward theoretical analysis, with antithetic termination enjoying strong performance guarantees. The new section will now include a full proof rather than just being presented as an empirical observation.

---

> > > > ### Author Response · Authors · 2023-08-18
> > > > **Any further questions?**
> > > >
> > > > Could the reviewer please kindly confirm whether the response above answers their remaining questions, or whether anything remains unclear?

---

> > > > ### Comment · Reviewer_SAEb · 2023-08-18
> > > > **Thanks for the response**
> > > >
> > > > > From 2-regularized Lap to d-regularized Lap
> > > >
> > > > Thanks for the explanation! I still have the following questions. I think one key usage of random feature based estimation of the kernel is that we do not need to estimate the entire kernel matrix $K_L^{(2)}$. For each entry (two nodes) of interest, we can estimate separately. However, based on the authors' response, it seems that we have to estimate the entire  $K_L^{(2)}$ and get $K_L^{(d)}$ via multiplication. I am not quite sure that such multiplication may have good variance. I saw the authors do some experiments shown in Table 2. But it is a comparison between GRF and q-GRF. I buy the results because q-GRF is more accurate than GRF. I am curious how q-GRF is compared with other ways to estimate $K_L^{(d)}$.  Regarding the multiplication idea, I am curious why not estimate $K_L^{(1)}$ (effective resistance) and do multiplication to get $K_L^{(2)}$ and $K_L^{(d)}$ in general. Why do we need $K_L^{(2)}$?

---

> > > > > ### Author Response · Authors · 2023-08-18
> > > > > **Thanks for the further questions**
> > > > >
> > > > > We thank the reviewer for their response and further interesting questions. We are pleased that they are convinced that q-GRFs outperform the previously introduced class of GRFs, which is the central claim of the paper. The question of how to generalise GRFs (and their q-GRF variants) to other graph kernels is not our chief focus, but since it is an interesting problem we are happy to discuss it in more detail below.
> > > > >
> > > > > Scalable estimation of graph kernels with random walks is a new field and we will be grateful if the reviewer could point us to any other specific methods for computing $\mathbf{K}_L^{(d)}$ (including $d=1$). Should they exist and use terminating random walks, we strongly suspect that antithetic termination will still prove an effective QMC mechanism. We would be happy to run further experiments and attempt theoretical analysis in the future, but we were unable to find other such methods in the literature.
> > > > >
> > > > > The reviewer is correct that, though unbiased, multiplying independent estimators of $\mathbf{K}_L^{(2)}$ to obtain estimators of $\mathbf{K}_L^{(d)}$ is likely not the most efficient approach. It is straightforward to calculate the variance: for example, $\textrm{Var} (\widehat{\mathbf{K}}_L^{(4)}) = \textrm{Var} (\widehat{\mathbf{K}}_L^{(2)})^2 + 2\textrm{Var} (\widehat{\mathbf{K}}_L^{(2)}) {\mathbf{K}_L^{(2)}}^2$. As the reviewer says, this will be suppressed with q-GRFs because we have proved that in this case $\textrm{Var} (\widehat{\mathbf{K}}_L^{(2)})$ is smaller. Since we derived closed forms for $\textrm{Var} (\widehat{\mathbf{K}}_L^{(2)})$ with both mechanisms it is trivial to explicitly obtain $\textrm{Var} (\widehat{\mathbf{K}}_L^{(d)})$, but even just inspecting the small errors in Table 2 which uses as many as $500$ such multiplications we can see that the approximation quality is still good, even with just $10$ walkers per node.
> > > > >
> > > > > That said, we agree that a method to *directly* estimate $\mathbf{K}_L^{(d)}$ without recourse to multiplying several estimators would be preferable. Adjusting the GRF mechanism to do this is nontrivial and a problem we intend to dedicate substantial time to. We have made some progress, at a high-level by introducing an *extra function of walker path length that modulates its load*, separate from the existing functions $\widetilde{\omega}(\omega)$ and $p(\omega)$. We intend to carry out a detailed theoretical and empirical exploration of these methods in the future; our strong intuition is that antithetic termination will once again prove very effective at reducing kernel estimator variance.
> > > > >
> > > > > We thank the reviewer for prompting this interesting discussion, but trust that they appreciate that the central contribution of the present manuscript is a novel QMC method for random walkers including for Choromanski's GRF mechanism, *not* a new GRF mechanism for direct estimation of different kernels. This is a substantial, open research problem and deserves to be properly addressed in a separate paper.
> > > > >
> > > > > To update the reviewer, we have also now proved that inducing antithetic termination suppresses the error on estimates of the PageRank vector for arbitrary graphs and arbitrary termination probabilites $p$. This surprisingly strong result is in the manuscript.

---

> > > > > > ### Author Response · Authors · 2023-08-20
> > > > > > **End of discussion period**
> > > > > >
> > > > > > As the discussion period draws to a close, we respectfully ask that, if satisfied with the responses above, the reviewer kindly considers raising their score. We again thank them for their interesting questions, to which we hope we have provided exhaustive answers.

---

### Official Review · Reviewer_wGeX · 2023-08-05

**Soundness:** 3 good
**Presentation:** 3 good
**Contribution:** 4 excellent
**Rating:** 9
**Confidence:** 4

**Summary:**

The paper introduces a method for graph random features (GRFs). Graph random features is the graph equivalent (i.e. kernels defined on graphs) of kernel matrix approximation using Random Features (Rahimi & Recht). Recently Chromanski et al. had proposed a method for GRF which uses a series of random walks.

In the Euclidean setting, the quasi-Monte Carlo variants of the Random Features methods have been developed since they provide better convergence properties. So the goal of this paper is to devise an algorithm for the quasi-Monte Carlo GRFs setting.

The idea is to correlates the length of the random walks by imposing 'antithetic termination' which is effectively a procedure to obtain a more diverse ensemble of graph random walks.

The paper provides several theoretic results related to the error guarantees as well as a brief empirical study.

**Strengths:**

+ Original idea with strong practical impact and wide ranging downstream applications as time-efficient approximation of the graph diffusion process is used in many scientific fields.
+ The paper is of very high quality: Well written and clear.
+ Good literature review and problem justification, as well as empirical analysis.



**Weaknesses:**

- Some typos/grammatical mistakes (esp. definite/indefinite articles), please proof read before publishing
- Some questions below as well.

**Questions:**

* It seems that all the experiments are on synthetic datasets. Can the authors confirm that or clarify why they didn't test this on at least on real dataset?

* Sec 4.3 cites the Dhillon paper regarding the kernel matrix they used but that paper has experiments on both handwritten image data and a gene dataset. It is not clear what dataset this experiment is based on. Please clarify this point.

* Additionally,  I would suggest adding a sentence in the text describing the nature of the datasets so that the reader doesn't have to go back to the original papers to understand the eval/test data. Similarly for Sec 4.4 what are the matrices used by Dawson-Haggerty, 2023? Please add a 1 sentence clarification.

**Limitations:**

The authors address the limitations and potential downstream impact in a dedicated section at the end.

---

> ### Author Rebuttal · Authors · 2023-08-08
>
> We thank the reviewer for their reading of the manuscript and positive comments. We are delighted that they recognise the scheme’s originality, strong practical impact and broad downstream applications. They note that the work is well-motivated and the manuscript is clear. We address their questions in detail below and will be happy to follow up with any further clarifications.
>
> 1. *Typos*: we thank the reviewer for flagging minor errors and will be sure to check the text again prior to submission of the final draft.
>
> 2. *Synthetic and real datasets*: we refer to a graph as ‘synthetic’ if it is generated according to some mathematical scheme (deterministic or stochastic) and ‘real-world’ if it is derived from actual data. By this definition, the experiments in sections 4.1 (direct estimation of the $2$-regularised Laplacian kernel) and 4.2 (simulation of graph diffusion) use an equal split of synthetic (2x ER, binary tree and ladder) and real-world (karate, dolphins, football, eurosis) graphs. Section 4.3 (clustering) uses exclusively real-world graphs, all of which are standard for benchmarking. Section 4.4 (kernel regression) uses triangular mesh graphs from a popular online repository, corresponding to a variety of 3D shapes such as a cylinder and a teapot. It is unclear whether such computer graphics should be considered synthetic or real but they are certainly standard. We hope this clarifies the split of synthetic and real data but will be happy to discuss further if we misinterpret the reviewer’s question.
>
> 3. *Relationship to Dhillon et al. (2004)*: we cite Dhillon’s paper because it provides a clear description of kernelised $k$-means clustering (Alg. 1 in Sec. 2.1, page 552). In our work, we implement this algorithm to cluster the nodes of $8$ real world graphs, comparing the results when we use the exact kernel (yielding the groundtruth clusters) and its approximation with i) GRFs and ii) q-GRFs. We find that, since q-GRFs provide a provably better approximation of the Gram matrix, they tend to return clusters closer to the groundtruth. We stress that all graphs considered correspond to real-world data and are of standard use in benchmarking. The reviewer correctly notes that Dhillon’s paper uses different datasets to us in experiments designed to elucidate the relationship between spectral clustering and kernelised $k$-means clustering, but these are not of direct relevance to our work.
>
> 4. *Clarification of nature of datasets, especially Dawson-Haggerty (2023)*: we thank the reviewer for their comment and direct them back to point 2 of our rebuttal. Dawson-Haggerty (2023) is a popular repository of triangular mesh graphs corresponding to 3D computer graphic objects, and here the task is to use kernel regression to predict the missing normal vector directions of the nodes. We will be sure to clarify this important point in the manuscript to save the reader from needing to follow the link to the repository.
>
> We again thank the reviewer for their attentive reading of the draft and excellent review.

---

> ### Comment · Reviewer_wGeX · 2023-08-17
> **thanks for your response**
>
> Thanks for your detailed response. It addresses my questions and given the questions from the other reviewers and the subsequent responses and discussions I am leaving my scores unchanged.

---

> > ### Author Response · Authors · 2023-08-17
> > **Thanks for your reply**
> >
> > We thank the reviewer for their reply, and again for their detailed reading of the text and strong review.

---

### Author Rebuttal · Authors · 2023-08-08

We are grateful for the reviewers’ careful readings of the manuscript and thoughtful feedback. We are delighted to receive such positive comments about the elegance of the mechanism and our exhaustive theoretical and empirical analysis. They also make several helpful suggestions which have further improved the draft, in particular prompting us to include a new section applying antithetic termination to PageRank (please see our individual responses for details and the attached PDF for results). This further showcases the strength and generality of the QMC approach even beyond graph kernel estimation. We address concerns and points of minor misunderstanding in the rebuttal and warmly encourage the reviewers to respond with any other queries.

---

### Decision · Program_Chairs · 2023-09-21

**Decision:**

Accept (spotlight)

**Comment:**

The reviewers were in consensus that the antithetic termination idea introduced in this paper is simple, novel, empirically effective, and may have further applications beyond those explored in the work. All felt that the paper was also well-written. Thus, the paper was judged to be a clear accept.